# The proteasome 19S cap and its ubiquitin receptors provide a versatile recognition platform for substrates

Kirby Martinez-Fonts[1], Caroline Davis [1], Takuya Tomita[1], Suzanne Elsasser[2], Andrew R. Nager[3], Yuan Shi[2], Daniel Finley[2]* & Andreas Matouschek [1]*

Proteins are targeted to the proteasome by the attachment of ubiquitin chains, which are markedly varied in structure. Three proteasome subunits–Rpn10, Rpn13, and Rpn1–can recognize ubiquitin chains. Here we report that proteins with single chains of K48-linked ubiquitin are targeted for degradation almost exclusively through binding to Rpn10. Rpn1 can act as a co-receptor with Rpn10 for K63 chains and for certain other chain types. Differences in targeting do not correlate with chain affinity to receptors. Surprisingly, in steady-state assays Rpn13 retarded degradation of various single-chain substrates. Substrates with multiple short ubiquitin chains can be presented for degradation by any of the known receptors, whereas those targeted to the proteasome through a ubiquitin-like domain are degraded most efficiently when bound by Rpn13 or Rpn1. Thus, the proteasome provides an unexpectedly versatile binding platform that can recognize substrates targeted for degradation by ubiquitin chains differing greatly in length and topology.

[1] Department of Molecular Biosciences, The University of Texas at Austin, Austin, TX 78712, USA. [2] Department of Cell Biology, Harvard Medical School, Boston, MA 02115, USA. [3] Department of Biology, Massachusetts Institute of Technology, Boston, MA 02139, USA. *email: daniel_finley@hms.harvard.edu; matouschek@austin.utexas.edu

Regulated protein degradation in eukaryotic cells is primarily mediated by the ubiquitin proteasome system. The protease at its center, the proteasome, is composed of a 19-subunit regulatory particle (RP), which recognizes substrates, and a 28-subunit core particle (CP), which degrades them. Substrates are passed from the RP to the CP through narrow channels within each assembly[1]. The proteasome degrades regulatory proteins, removes misfolded and damaged proteins, and digests foreign proteins as part of the adaptive immune system[2,3].

Proteins are targeted to the proteasome primarily through the attachment of polyubiquitin chains. The canonical targeting signal is a chain of at least four ubiquitin molecules linked to each other through isopeptide bonds between the C terminus of one ubiquitin and lysine 48 (K48) of the next[4,5]. These polyubiquitin chains are typically attached to lysine residues in the target protein. Shorter chains and single ubiquitin molecules (mono-ubiquitin) can also mediate degradation, especially when several are attached to the same target protein[6–8].

Once a ubiquitinated protein is bound, the proteasome initiates its translocation through the RP into the CP at an unstructured region in this protein[9,10]. The location of the ubiquitin chain relative to an initiation site conditions whether the protein can be degraded[11]. The two sites must be at an appropriate distance from one another, presumably because the proteasome has to simultaneously bind the ubiquitin chain via the ubiquitin receptors and initiate degradation at an unstructured region via another receptor. The receptor for the initiation region is most likely within the axial channel of the heterohexameric Rpt ring of ATPases, located at the heart of the RP. The substrate is subsequently driven through the channel into the CP by ATP hydrolysis. The amino acid sequence of the initiation region in the substrate also affects degradation[12–15].

Ubiquitin chains can be linked not only through K48 but also through any of the six other lysines within ubiquitin, as well as its N-terminal amine at M1, with the most common linkage sites being K48, K63, and K11 (ref. [16]). Some of these chains promote proteasomal degradation[16–18] whereas others are involved in other cellular processes[19,20]. For example, K63-linked ubiquitin chains are mostly associated with endocytosis, translation, autophagic targeting, signaling in innate immunity pathways, and DNA repair[21]. However, K63-linked chains can also target proteins for degradation in vitro[22–24] and at least under some circumstances in vivo[16,23,25–28]. In addition, a ubiquitin-ligase associated with the yeast proteasome, Hul5, enhances stress-inducible protein degradation as well as the processivity of degradation of a variety of proteasome substrates through the synthesis of K63-linked ubiquitin chains[29–32]. K11-linked ubiquitin chains have been shown to target for proteasomal degradation in the ERAD (Endoplasmic Reticulum-Associated [protein] Degradation) pathway and in cell cycle progression through the destruction of cell cycle regulators[16,17,33]. K11 linkages are often found in mixed and branched ubiquitin chains, and it is unclear if they can target proteins to the proteasome as homotypic K11-linked chains[34–36].

The stoichiometric proteasome subunits Rpn1, Rpn10, and Rpn13 serve as ubiquitin receptors[37–40]. Other subunits, such as Rpt5 (ref. [41]) and Sem1/Dss1 (ref. [42]) have also been proposed to bind ubiquitin but are less well characterized and may not recognize ubiquitin when assembled in the complete proteasome[40]. Rpn10 is located closest to the substrate translocation channel of the Rpt ring whereas Rpn13 is located at the top of the RP, somewhat further from substrate entry port, and Rpn1 is located on the opposite side of the degradation channel relative to Rpn10 (Fig. 1). Rpn10 binds to ubiquitin chains through UIM (Ubiquitin Interacting Motif) domains[37,43,44], which consist of single α-helices and are flexibly linked to an N-terminal von

Willebrand factor A (VWA) domain docked tightly into the proteasome structure; Rpn13 binds ubiquitin chains with a PRU (Pleckstrin-like Receptor for Ubiquitin) domain, which is docked directly into the proteasome RP and interacts with ubiquitin through three loops[38,39]; and finally, Rpn1 binds ubiquitin in two grooves flanked by α-helices in its toroid repeat region, which is an integral part of the RP[40].

All three of the proteasomal ubiquitin receptors can also bind substrates indirectly by serving as receptors for ubiquitin-like (UBL) domains of UBL-UBA proteins[38–40,45–48]; hence, we refer to them below as Ub/UBL receptors. UBL-UBA proteins are thought to function as diffusible substrate receptors by binding to the proteasome and to ubiquitinated proteins[49–57]. Rpn1 also binds the UBL of the deubiquitinating enzyme (DUB) Ubp6, though at a separate site called the T2 site[40,47,58,59].

The presence of multiple Ub/UBL receptors on the proteasome raises interesting questions and we do not understand how the receptors cooperate in protein degradation. Why does the proteasome have multiple ubiquitin receptors? Do they function in multivalent recognition pathways or individually? Does each receptor have certain preferences for specific linkages, or do they recognize ubiquitin chains of different sizes or topologies? Do different receptors recognize substrates of different conformations? We address such questions here by characterizing the degradation of well-defined substrate proteins with specific arrangements of ubiquitin chains and using purified proteasomes in which individual ubiquitin receptors have been mutated to attenuate target recognition. We find that the ensemble of ubiquitin receptors on the proteasome provides a versatile interaction platform, allowing the recognition of substrates with different conformations and ubiquitin chains of different length and linkages. Rpn10 functions as the primary ubiquitin chain receptor and Rpn13 and Rpn1 can cooperate with Rpn10 to enhance degradation of some proteins. The most robust degradation signals may not be long ubiquitin chains but multiple chains.

## Results

**Proteasome substrates with defined polyubiquitin chains**. We created substrates with ubiquitin chains of defined lengths and linkages attached to a base protein[24] (Supplementary Fig. 1). The base protein was built around a central fluorescent protein domain, which was flanked by a ubiquitin domain at its N- or C terminus and a disordered stretch of amino acids at the opposite end (Fig. 1a). The fluorescent domain was a circular permutant of superfolder GFP in which the wild-type N- and C-termini were connected by a short linker and a new N terminus created at the beginning of the ninth β-strand of the wild-type protein. The circular permutant was biochemically well behaved, fluorescent, and readily degraded by the proteasome[24,60]. The disordered region was a sequence of 35 or 95 amino acids derived from *S. cerevisiae* cytochrome $b_2$, either of which allows the proteasome to initiate degradation of the substrate effectively[11,12]. The N- and C-termini of GFP (and of the circular permutant) are adjacent to each other so that ubiquitin chain and initiation region are close to each other in space. The co-translated ubiquitin domain served as the attachment point for polyubiquitin chains[24]. It was mutated at position 76 (G76V) to prevent its cleavage from GFP by ubiquitin C-terminal hydrolases. In addition, we generated a substrate that contained two ubiquitin domains near the N terminus, separated by a 35 amino acid long linker, to attach two ubiquitin chains to a protein. Finally, we created a set of proteins that contained the ubiquitin-like domain from Rad23 (UBL) fused to their N terminus instead of a ubiquitin domain[11].

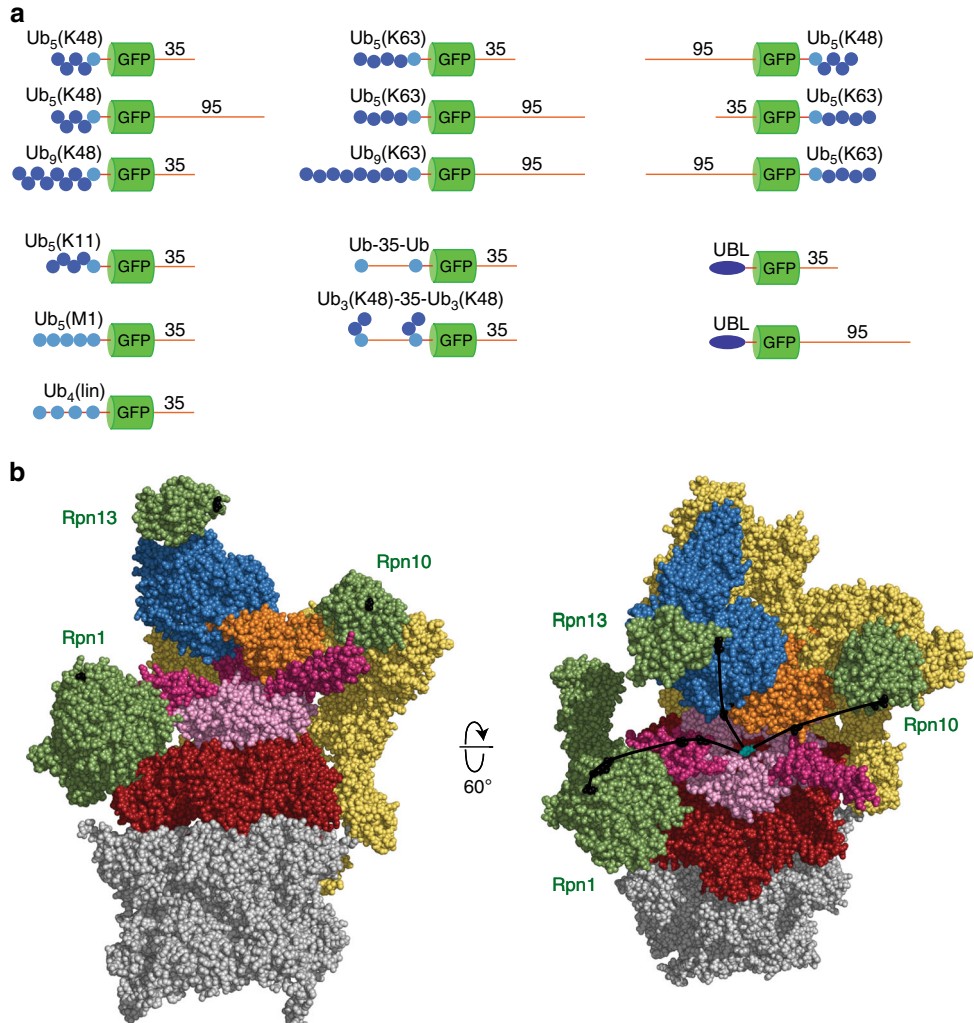

**Fig. 1 Model substrates and proteasome particles. a** Schematic representation of the substrate proteins analyzed. GFP shown in green, co-translated ubiquitin domains shown in light blue, enzymatically added ubiquitin moieties shown in dark blue, intrinsically unfolded regions of proteins shown as red lines. In the three-dimensional structure of GFP, the N- and C-termini are adjacent to each other so that ubiquitin chain or UBL and initiation region are next to each other in space. **b** Structure of the proteasome in the s1 state (PDB4CR2) in two orientations. The structure broadly includes one half of the 20S core particle (gray) and the 19S regulatory particle (multiple colors). The ATPase (Rpt) subunits are color-coded respect to domains rather than subunits, with the AAA+ domains in magenta, the OB ring in pink, and the coiled-coil domains in purple. The position of the pore-1 loop is indicated by a red circle in the right panel. The Ub/UBL receptors Rpn1, Rpn10, and Rpn13 are shown in green, and Rpn2 is shown in light blue. The DUB Rpn11 is shown in yellow, and the remaining components of the lid are shown in light yellow. Black lines indicate a direct path along the surface of the proteasome from the Ub/UBL-binding sites in Rpn1 and Rpn13 to the pore-1 loop at the top of the ATPase ring. The Ub/UBL-binding UIM domain of Rpn10 is not visualized in this structure; instead, the black line indicates the path from the last resolved residue of the VWA domain of Rpn10 to pore-1 loop. The distances are approximately 107 Å for Rpn13, 100 Å for Rpn1, and 95 Å for Rpn10. The UIM domain of Rpn10 is attached to the VWA domain by an unstructured linker of approximately 20 amino acids, which may reduce the distance from the ubiquitin-binding site to the pore-1 loop by 20–30 Å.

**Proteasomes with defined Ub/UBL receptors**. We purified yeast proteasome particles in which individual receptors were inactivated by amino acid substitutions in their ubiquitin-binding surface (Fig. 1b). Mutations in the Ub/UBL receptors can affect the levels of co-purified proteasome-interacting proteins such as Ubp6 and Rad23 (refs. [40,61]). To remove this variable, we purified the RP and the CP separately through a 3xFLAG affinity tag on Rpn11 and Pre1, respectively, using high-salt washes to strip off proteasome-interacting proteins[40,61]. We then reconstituted the proteasome at a molar ratio of 1:2 of CP:RP (Supplementary Fig. 2, Supplementary Note 1).

We define the wild-type proteasome as a reconstituted proteasome particle bearing wild-type Ub/UBL receptors. Rpn1 was mutated in its T1 toroid region to abolish binding of the UBL domain of Rad23 and to greatly decrease ubiquitin binding as described (*rpn1-ARR*[40]); Rpn10 was mutated in its UIM domain to reduce ubiquitin and UBL binding to a similar extent as deleting the entire UIM (*rpn10-uim*[43,55]); and Rpn13 was mutated in its PRU domain to abolish ubiquitin binding and greatly decrease UBL binding (*rpn13-pru*[40]). We refer to particles by indicating which ubiquitin receptors are intact: particles harboring the *rpn1-ARR* and *rpn13-pru* mutations are Rpn10 proteasomes, particles harboring the *rpn1-ARR* and *rpn10-UIM* mutations Rpn13 proteasomes, and particles harboring the *rpn10-UIM* and *rpn13-pru* mutations Rpn1 proteasomes (Supplementary Table 1). Proteasomes harboring point mutations in only one receptor are similarly named according to the unaffected receptors (i.e., Rpn10/13 harbors the *rpn1-ARR* mutation); finally,

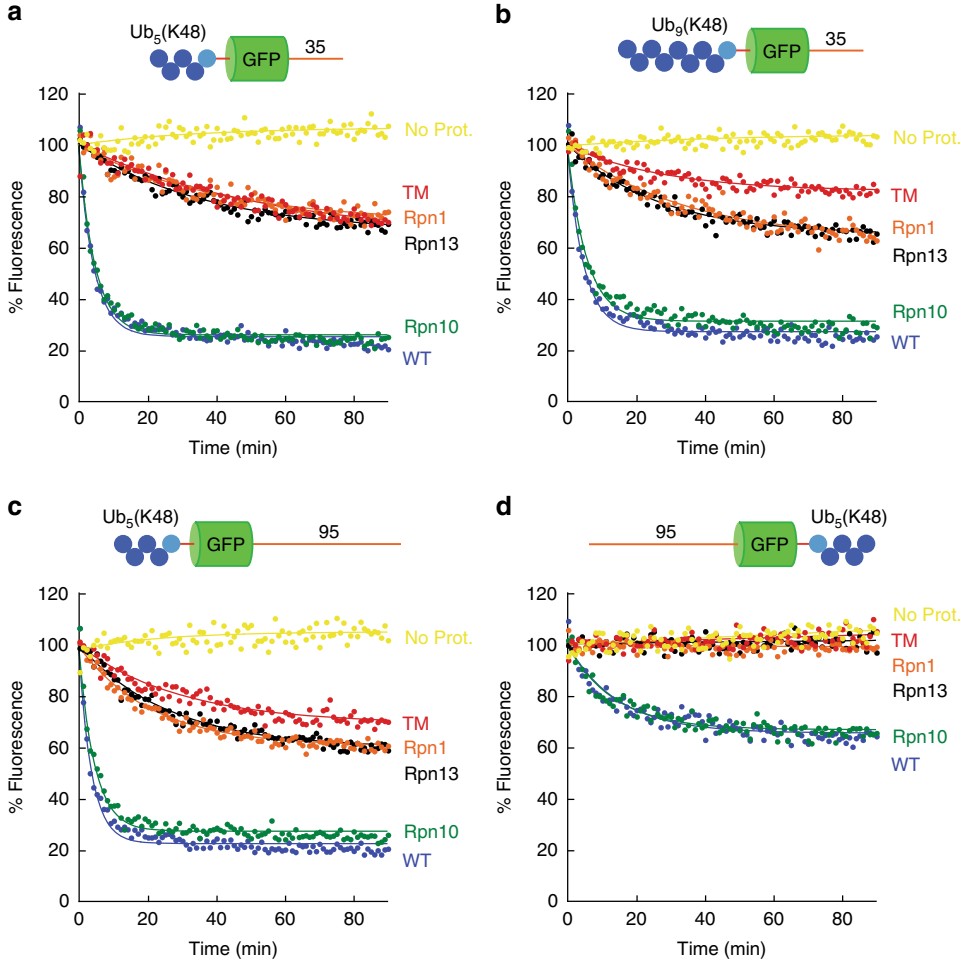

**Fig. 2 Degradation of substrates with K48-linked polyubiquitin chains.** Degradation of substrate proteins with K48-linked ubiquitin chains by the indicated proteasome mutants was followed under single-turnover conditions (5 nM substrate, 25 nM proteasome) in the presence of 1 mM ATP at 30 °C. The graphs show substrate fluorescence as a percentage of the initial fluorescence as a function of time in minutes. Proteasome types are described in Supplementary Table 1 (TM triple mutant proteasome). Each panel shows the degradation of particular substrates, and as follows: **a** $Ub_5$(K48)-GFP-35; **b** $Ub_9$(K48)-GFP-35; **c** $Ub_5$(K48)-GFP-95, and **d** 95-GFP-$Ub_5$(K48). Source data are provided as a Source Data file.

proteasomes in which Rpn1, Rpn10, and Rpn13 all carry substitutions are referred to as TM (triple mutant) proteasomes.

**Substrates with K48-linked ubiquitin chains.** We first asked how the proteasome recognizes proteins targeted for proteolysis by the canonical degradation signal as defined originally by Pickart and colleagues[5], which is formed by a polyubiquitin chain consisting of four or more ubiquitin molecules linked through K48. We attached chains of four or eight ubiquitin molecules to base proteins (i.e., the substrate protein as translated consisting of a ubiquitin domain, the GFP domain, and the initiation region) near either their N- or C-termini as described previously[24]. We then tested the degradation of these proteins by mutant proteasomes in excess over substrate to ensure single-turnover conditions (Fig. 2). All four substrates were degraded rapidly by wild-type proteasomes, though the substrate with a ubiquitin chain at its C terminus was degraded less effectively (Fig. 2c, d, Supplementary Table 2).

Degradation for these substrates with K48-linked chains was mediated by Rpn10. Attenuating ubiquitin binding by Rpn13 and Rpn1 by mutation did not affect degradation of any of these four substrates significantly, and Rpn10 proteasome degraded the proteins as effectively as wild-type proteasome (Fig. 2, Supplementary Table 2). Rpn1 and Rpn13 can contribute to degradation

of larger proteasome substrates because Rpn1 proteasome and Rpn13 proteasome degraded $Ub_9$(K48)-GFP-35 and $Ub_5$(K48)-GFP-95 somewhat better than TM proteasome (Fig. 2b, c, Supplementary Table 2).

Mutating all the three known ubiquitin receptors simultaneously (TM proteasome) inhibited degradation substantially but not always completely (Fig. 2a–c, Supplementary Table 2). Only degradation of the substrate with a C-terminal ubiquitin chain was prevented entirely in TM proteasome (Fig. 2d); all three substrates with N-terminal ubiquitin chains were degraded even when all known ubiquitin receptors were mutated, albeit at a reduced rate. The residual degradation was not due to contaminating proteases or photobleaching, as substrate fluorescence remained constant over time in the absence of ATP or proteasome (Fig. 2, Supplementary Fig. 3). Degradation depended on the ubiquitin chain, as unmodified GFP-35 remained stable in the same assay (Supplementary Fig. 4). Hence, either the mutations introduced in the ubiquitin receptors do not abolish ubiquitin binding completely or, more likely, the proteasome contains an as yet unidentified ubiquitin receptor.

**Substrates with K63-linked ubiquitin chains.** The second most abundant ubiquitin–ubiquitin linkage in yeast and human cells is through K63. K63-linked chains can also target proteins for

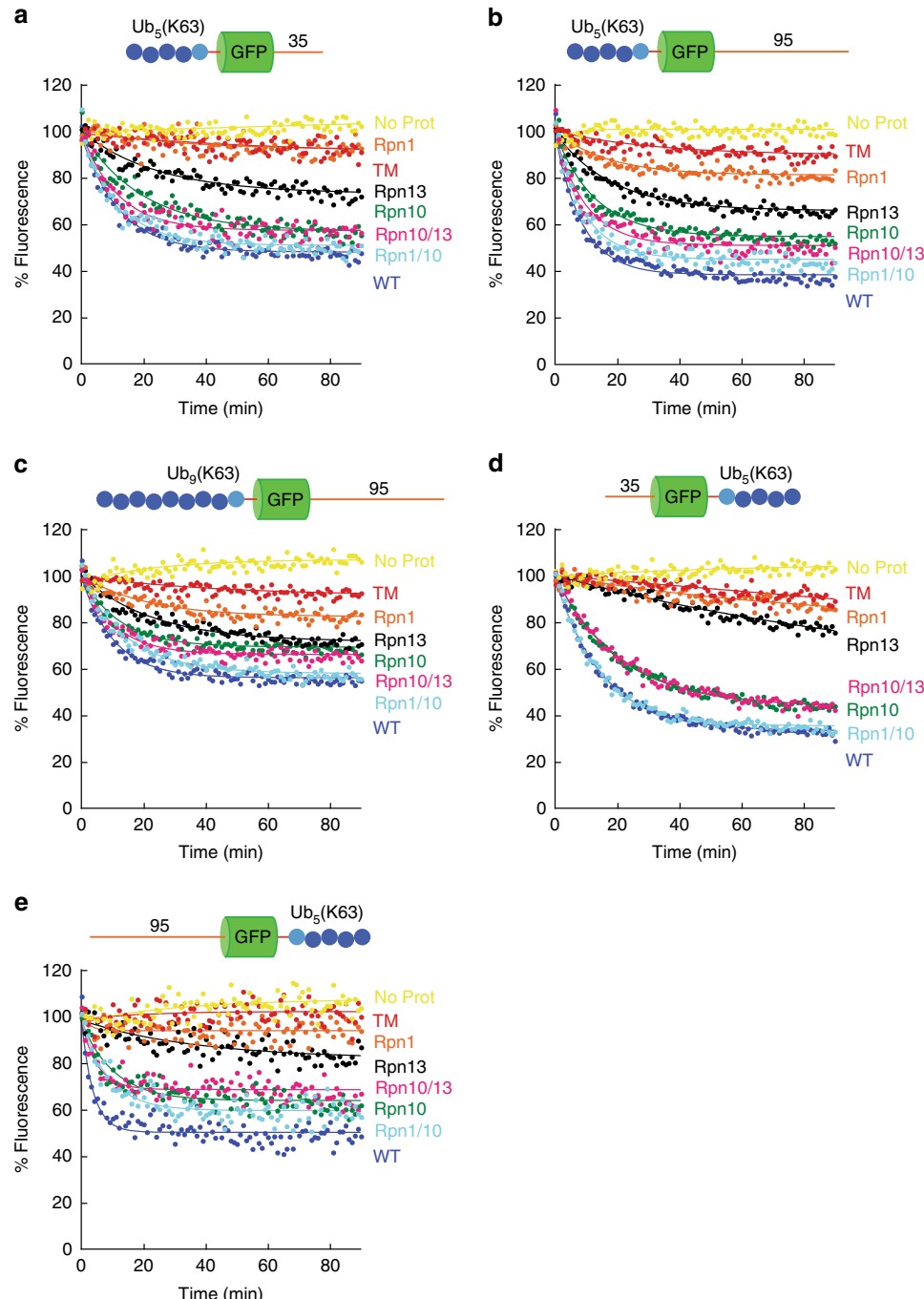

**Fig. 3 Degradation of substrates with K63-linked polyubiquitin chains.** Degradation of substrate proteins with K63-linked ubiquitin chains by the indicated proteasome mutants was followed under single-turnover conditions (5 nM substrate, 25 nM proteasome) in the presence of 1 mM ATP at 30 °C. The graphs show substrate fluorescence as a percentage of the initial fluorescence as a function of time in minutes. Proteasome types are described in Supplementary Table 1 (TM triple mutant proteasome). Each panel shows the degradation of particular substrates, and as follows: **a** Ub$_5$(K63)-GFP-35; **b** Ub$_5$(K63)-GFP-95; **c** Ub$_9$(K48)-GFP-95; **d** 35-GFP-Ub$_5$(K63); and **e** 95-GFP-Ub$_5$(K63). Source data are provided as a Source Data file.

proteasomal degradation[22,23,35] though they are usually associated with other cellular processes[21] and generally not thought to target proteins to degradation in vivo[16,26]. Therefore, we asked how defined substrates with K63-linked ubiquitin chains are recognized and degraded by the proteasome.

Wild-type proteasome degraded all substrates with K63 chains (Fig. 3). Degradation was again mediated primarily by Rpn10, though the presence of other receptors enhanced degradation (Fig. 3, Supplementary Table 2). Rpn13 proteasome was also able to degrade these substrates but did so more slowly than Rpn10

proteasome. Rpn1 proteasome degraded the substrates even more slowly than Rpn13 though still faster than TM proteasome. Instead, Rpn1 functioned as a co-receptor with Rpn10, and Rpn1/10 proteasome degraded K63-chain-bearing substrates almost as well as wild-type proteasome. Interestingly, restoring the Rpn13-binding site in the Rpn10 proteasome (Rpn10/13 proteasome) did not have a substantial effect. Increasing the length of the K63 chains does not enhance degradation and may even attenuate it slightly [compare Ub$_5$(K63)- and Ub$_9$(K63)-GFP-95; Fig. 3b, c, Supplementary Table 2]. Mutating all three ubiquitin receptors

simultaneously abolished degradation almost completely, with TM proteasome degrading these proteins only slightly above background.

The proteasome degraded substrates with K63-linked ubiquitin chains somewhat more slowly than the best substrate with K48-linked chains. However, when attached near the C-termini, K63-linked chains were better degradation signals than K48-linked chains (Supplementary Fig. 5, Supplementary Table 2). Indeed, switching the ubiquitin chain attachment point from near the N terminus to near the C terminus of the protein slowed degradation for proteins with K48-linked chains but accelerated degradation for proteins with K63-linked chains. This observation suggests that the chain linkage type can influence substrate degradation rates by causing proteins to be presented to the proteasome in different orientations.

**Substrates with K11-linked and linear ubiquitin chains.** Ubiquitin linkages through K11 are found in both yeast and mammalian cells[16] though they may be rare in yeast[62]. They have been firmly linked to protein degradation as part of cell cycle regulation and ERAD[16,35,36,63]. However, our model protein with a K11-linked ubiquitin chain [Ub$_5$(K11)-GFP-35] was not degraded by any variant of the proteasome (Fig. 4a) and persisted in the ubiquitinated form (Supplementary Fig. 6), consistent with other in vitro studies[24,34].

Linear ubiquitin chains, in which the ubiquitin molecules are attached directly to the each other through peptide bonds between the N terminus (i.e., M1) of one ubiquitin molecule and the C terminus of the next ubiquitin molecule, are synthesized by the E3 enzyme LUBAC in the NFκB signaling pathway[64]. Like K11 chains, linear chains failed to target the base protein for degradation [Ub$_5$(M1)-GFP-35, Fig. 4b] and the protein persisted in the ubiquitinated form (Supplementary Fig. 6). LUBAC activity has not been observed in yeast and so we performed equivalent degradation experiments with proteasome particles purified from mouse embryonic fibroblasts expressing Rpn11 with a C-terminal Flag-tag[65]. Degradation of the various substrates by mouse proteasome followed the same pattern as degradation by yeast proteasome (Supplementary Fig. 7). In particular, substrate modified by a ubiquitin chain linked through Lys11 near its N terminus is not degraded by mouse proteasome, while substrate with a Lys48-linked chain near its N terminus, and to a lesser extent substrate with a Lys63-linked chain, is degraded (Supplementary Fig. 7).

Interestingly, inserting a short linker (GSGGGG) between the ubiquitin domains converted the fusion of four ubiquitin domains into an effective proteasome targeting signal[9,66–68], which can be recognized by Rpn10 or Rpn13 [Ub$_4$(lin)-GFP-35; Fig. 4c]. These substrates are easy to synthesize and thus serve as a convenient tool to investigate ubiquitin-dependent proteasome activity. The degradation profile of these substrates was similar to that of proteins carrying a K63 chain [compare Ub$_5$(K63)-GFP-35 in Fig. 3a to Ub$_4$(lin)-GFP-35 in Fig. 4c], and may derive from analogous structural features of the K63-linked modifications and Ub$_4$(lin). Ub$_4$(lin)-GFP-35 may also mimic a protein with four monoubiquitin modifications (see below).

**Proteasome affinity does not correlate with degradation.** A simple explanation for different contributions of the three Ub/UBL receptors to proteasomal degradation would be that the rate of degradation via a receptor correlates with its affinity for the ubiquitin chains. To estimate these affinities in the context of the proteasome, we measured the ability of free chains to inhibit the degradation of substrates competitively (Fig. 5, Supplementary Fig. 8). For wild type and Rpn10 proteasome, we performed

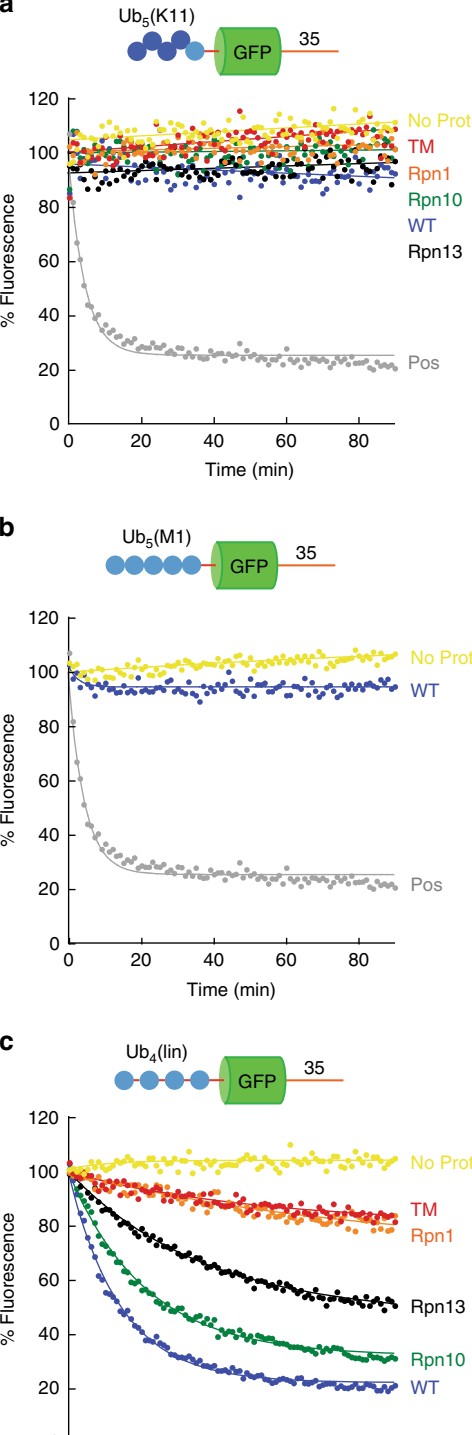

**Fig. 4 Degradation of substrates with K11-linked, M1-linked, and linear polyubiquitin chains.** Degradation of substrate proteins with the shown ubiquitin chains by the indicated proteasome mutants was followed under single-turnover conditions (5 nM substrate, 25 nM proteasome) in the presence of 1 mM ATP at 30 °C. The graphs show substrate fluorescence as a percentage of the initial fluorescence as a function of time in minutes. Proteasome types are described in Supplementary Table 1. Each panel shows the degradation of particular substrates, and as follows: **a** Ub$_5$(K11)-GFP-35; **b** Ub$_5$(M1)-GFP-35, and **c** Ub$_4$(lin)-GFP-35 by mutant proteasome as indicated. For panels **a** and **b**, the positive control (Pos) is degradation of Ub$_5$(K48)-GFP-35 by wild-type proteasome. Source data are provided as a Source Data file.

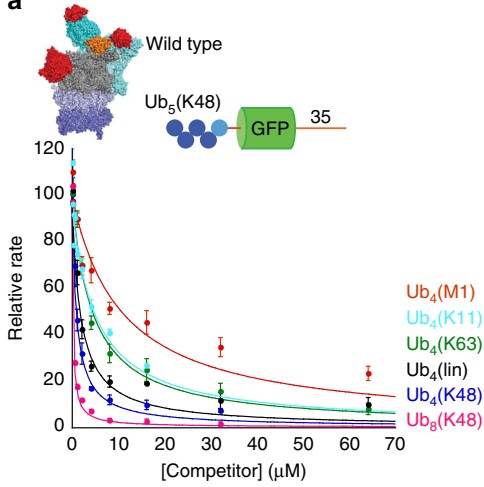

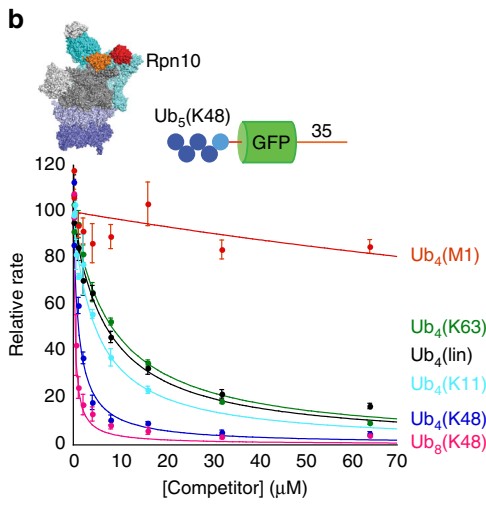

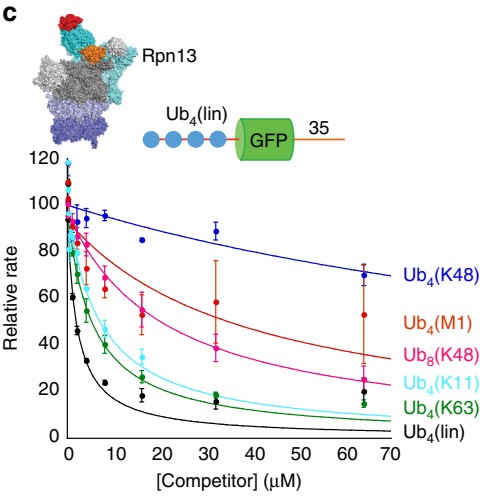

**Fig. 5 Inhibition of substrate degradation by free ubiquitin chains.**
Degradation of substrate proteins with three different ubiquitin chain modifications by the indicated proteasome types was followed under single-turnover conditions (5 nM substrate, 25 nM) in the presence of 1 mM ATP at 30 °C by measuring GFP fluorescence. Increasing concentrations of ubiquitin chains were added and the initial rates of the reactions determined by curve fitting. Graphs show initial rates of degradation as a percentage of the rate in the absence of competitor. **a** $Ub_5(K48)$-GFP-35 on WT proteasome, **b** $Ub_5(K48)$-Ub-GFP-35 on Rpn10 proteasome, **c** $Ub_4(lin)$-GFP-35 on Rpn13 proteasome inhibited by free ubiquitin chains. Error bars show standard errors derived from at least three replicate experiments. Source data are provided as a Source Data file.

with roughly the same apparent affinity as wild-type proteasome (Supplementary Table 3), and the two proteasomes degraded substrates with K48-linked chains equally well (Fig. 2a). K63-linked chains bound Rpn10 proteasome half as well as wild-type proteasome, again closely fitting the relative rates of degradation of substrates targeted by these chains (Supplementary Tables 2 and 3). Thus, for some substrates and proteasome variants, the initial rate of degradation in single-turnover experiments correlated with the affinity of the free ubiquitin chains to the proteasome. However, for other substrate proteins the relationship breaks down. Substrates with K63-linked chains near the C terminus were degraded faster than substrates with K48-linked chains near the C terminus (Figs. 2 and 3, Supplementary Fig. 4, Supplementary Table 2), despite the higher proteasome affinity of K48 chains. K48- and K63-linked chains near their N terminus were degraded equally well by Rpn13 proteasomes, despite the observation that free K63 chains bind ~30-fold better than free K48 chains. The proteasome bound K11-linked chains with essentially the same affinity as K63-linked chains (Fig. 5, Supplementary Table 3) but degraded only proteins modified by K63 chains (Fig. 4a). Similarly, M1-linked ubiquitin molecules (i.e., ubiquitin chains linked N to C terminus, $Ub_4(M1)$) bound to the proteasome in competition experiments (Fig. 5) but did not target proteins to degradation (Fig. 4b). Thus, for both K11-linked and M1-linked ubiquitin chains, the failure to target proteins for degradation cannot be explained by low affinity for the proteasome.

In conclusion, protein affinity to the proteasome by itself does not predict degradation. Presumably, the placement of the substrate on the proteasome affects degradation either by determining how well the proteasome is able to engage the protein at the initiation site or by inducing the proteasome to take up more (or less) active conformations.

**Steady-state analysis of degradation kinetics.** To investigate further how the proteasome discriminates among different substrates, we analyzed degradation using steady-state kinetics. Analysis of degradation rates at different substrate concentrations (i.e., Michaelis–Menten analysis) yields a specificity constant, $k_{cat}/K_M$, which reveals how a particular proteasome variant would discriminate between different substrates were they to compete with each other (Fig. 6, Supplementary Table 4). Steady-state kinetics also provide parameters that report on the maximal turnover number when the proteasome is saturated with substrate ($k_{cat}$) and the apparent affinity of substrate and reaction intermediates to the proteasome ($K_M$). The results of these experiments were largely in agreement with the conclusions of the single-turnover experiments; proteins with K48-linked ubiquitin chains near their N terminus were better substrates of wild-type proteasome than proteins with K63 chains at the same position (Fig. 6, Supplementary Table 4; i.e., had higher $k_{cat}/K_M$ values).

the competition experiments with $Ub_5(K48)$-GFP-35 as the substrate. For Rpn13 proteasome we instead used $Ub_4(lin)$-GFP-35, because it degrades $Ub_5(K48)$-GFP-35 only poorly (Fig. 4c).

K48-linked ubiquitin chains inhibited degradation roughly four times more effectively than K63-linked chains (Fig. 5, Supplementary Table 3), congruent with the fourfold difference in initial rate of degradation by the proteasome of substrates modified with these chains near their N-termini (Figs. 2 and 3, Supplementary Table 2). Rpn10 proteasome recognized free K48-linked chains

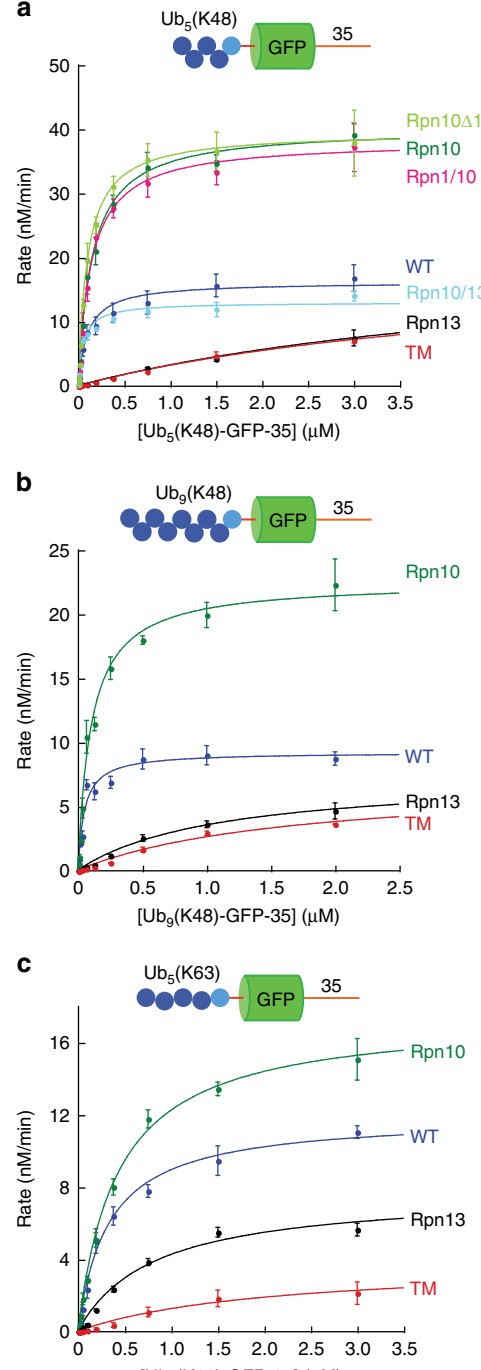

**Fig. 6 Michaelis–Menten analysis of proteasomal degradation.**
Degradation of substrate proteins with three different ubiquitin chain modifications by the indicated proteasome types was measured at 30 °C in the presence of 1 mM ATP and monitored by measuring GFP fluorescence. The indicated concentrations of substrate were incubated with 25 nM proteasome, fluorescence was recorded every minute for 120 min, and initial degradation rates were determined by curve fitting. The graphs show the initial rates as a function of substrate concentration. **a** $Ub_5(K48)$-GFP-35; **b** $Ub_9(K48)$-GFP-35; **c** $Ub_5(K63)$-GFP-35 by mutant proteasome. Error bars show standard errors derived from at least three replicate experiments. Source data are provided as a Source Data file.

The most notable feature of the steady-state analysis is that deletion of Rpn13 accelerated degradation when proteasomes were saturated with substrate and Rpn10 proteasomes were faster than wild type for all substrates (Fig. 6). In contrast, in single-turnover studies, wild-type proteasomes were equivalent to Rpn10 proteasomes or faster (Figs. 2 and 3). In particular, wild-type proteasomes degraded the substrate $Ub_5(K63)$-GFP-35 more rapidly than Rpn10 proteasomes under single-turnover conditions, but when saturated with substrate, Rpn10 proteasomes were faster than wild-type proteasomes. Thus, wild-type proteasomes are slower than mutant proteasomes in subsequent rounds of degradation, and a step after the unfolding of the GFP domain limits the overall degradation rates. For these substrates a step such as translocation, proteolysis, or product release is slower than unfolding. Since this effect is linked to mutation of Rpn13, it is possible that release of the ubiquitin chain from the receptor is limiting. Rpn11 releases ubiquitin chains from substrates, primarily cutting at the base of a ubiquitin chain, producing free chains that are comparable to the chains used in the competition experiments, which indeed inhibit degradation (Fig. 5).

We found that longer ($Ub_9$) chains were less effective than short ($Ub_5$) chains in targeting proteins for degradation in steady-state experiments despite the higher proteasome affinity of free longer chains (Supplementary Table 3). Thus, $Ub_5$ chains targeted GFP degradation with a $k_{cat}$ eightfold greater than $Ub_9$ chains. $Ub_5$ and $Ub_9$ chains performed equivalently in single-turnover studies (Supplementary Table 2), suggesting that slow product release may largely explain the $k_{cat}$ effect. Interestingly, the $k_{cat}$ effect is mostly linked to Rpn13, as attenuating ubiquitin binding by this receptor accelerates multiple turnovers.

Degradation by wild-type proteasome was attenuated by the Rpn13 receptor, despite Rpn13's low affinity for K48 ubiquitin chains by itself. Presumably the contribution of Rpn13 reflects avid recognition in conjunction with Rpn10, and mutants lacking Rpn13 binding probably release chains more quickly. Binding to Rpn13 may also alter the position of the substrate protein on the proteasome, reducing its chance to become engaged productively. Finally, Rpn13 moves relative to the rest of the proteasome during the ATPase cycle[1] so that constraining its position by substrate binding to Rpn13 and other parts of the proteasome simultaneously may inhibit the reaction cycle.

**Multiple ubiquitin chain are degraded efficiently.** An important development over recent years has been the realization that proteins can be targeted for proteasomal degradation through modification by multiple short ubiquitin chains or even multiple single ubiquitin molecules[6–8,69,70]. Individual ubiquitin receptors on the yeast proteasome bind either single ubiquitin moieties or the interface between two ubiquitin moieties within a single polyubiquitin chain[38,40,71]. Thus, binding of multiple ubiquitin chains simultaneously probably requires more than one receptor.

To explore this model, we constructed a substrate with two attachment sites for polyubiquitin chains by fusing a second ubiquitin domain to the N terminus of our base substrate. A 35 amino acid long linker provided separation between the two ubiquitin domains in the construct Ub-35-Ub-GFP-35. We then attached diubiquitin chains linked through K48 [$Ub_2(K48)$] to each ubiquitin domain in the base protein. This two-chain substrate [$Ub_3(K48)$-35-$Ub_3(K48)$-GFP-35, Fig. 7a] was degraded essentially as well as a protein with a canonical degradation signal of a single chain of four or more ubiquitin molecules [$Ub_5(K48)$-GFP-35, Fig. 2a or $Ub_9(K48)$-GFP-35, Fig. 2b] (Supplementary Table 2). However, the receptor specificity of this substrate was

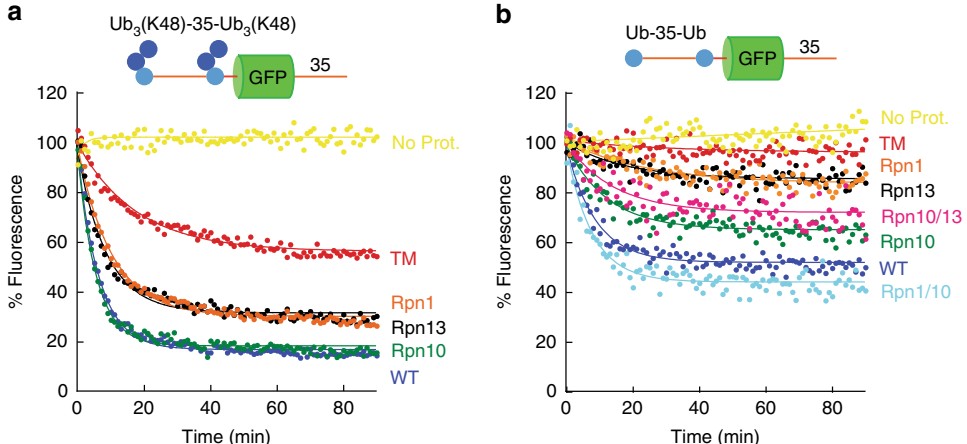

**Fig. 7 Degradation of multi-ubiquitinated substrates.** Degradation of substrate proteins modified with two K48-linked ubiquitin chains or bearing two co-translated single ubiquitin domains by the indicated proteasome types was followed under single-turnover conditions (5 nM substrate, 25 nM proteasome) in the presence of 1 mM ATP at 30 °C. The graphs show substrate fluorescence as a percentage of the initial fluorescence as a function of time in minutes. Proteasome types are described in Supplementary Table 1 (TM triple mutant proteasome). Each panel shows the degradation of particular substrates, and as follows: **a** Ub$_3$(K48)-35-Ub$_3$(K48)-GFP-35 and **b** Ub-35-Ub-GFP-35. Source data are provided as a Source Data file.

dramatically different from that of the single-chain substrate. Ub$_3$(K48)-35-Ub$_3$(K48)-GFP-35 was degraded best by wild type and Rpn10 proteasomes, but Rpn13 and Rpn1 proteasomes degraded it still about 50% as efficiently (Fig. 7a, Supplementary Table 2). The protein was degraded robustly even when all three receptors (TM proteasome) were mutated (Fig. 6a, Supplementary Table 2). These observations again suggest the presence of an unidentified receptor in the proteasome, though residual binding to one of the mutated receptors cannot be excluded.

To mimic a protein with two monoubiquitin tags, we analyzed degradation of an unmodified substrate containing only the two ubiquitin domains encoded in the polypeptide chain (Ub-35-Ub-GFP-35). This protein was degraded well by wild-type proteasome (Fig. 6b, Supplementary Table 2) and, remarkably, this two-ubiquitin signal was more effective than the classical degradation signal at the C terminus of GFP [35-GFP-Ub$_5$(K48), Fig. 2d]. Degradation of Ub-35-Ub-GFP-35 was almost completely prevented by mutating residues in the three known ubiquitin receptors. Rpn10 was again the most critical receptor for degradation but adding Rpn1 improved degradation twofold. On the other hand, adding Rpn13 to Rpn10 did not enhance degradation and may even have inhibited it slightly (Fig. 6b, Supplementary Table 2). Thus, we conclude that Rpn1 can function effectively as a co-receptor with Rpn10 for multiple ubiquitin tags in the same protein and enhance the degradation of multi-monoubiquitinated substrates.

**UBL receptor.** Substrates may be delivered to the proteasome by adaptor proteins that contain ubiquitin-like (UBL) domains, which can be recognized by all three ubiquitin receptors[38,40,48]. To investigate protein degradation mediated by UBL domains, we fused the UBL domain of Rad23 directly to our substrates by replacing their ubiquitin domain with the first 80 amino acids of Rad23 (UBL-GFP-35).

Rpn13 proteasomes degraded UBL-GFP-35 most efficiently, and Rpn10 proteasomes somewhat less so. In contrast, Rpn1 proteasome barely degraded this substrate and, strikingly, wild-type proteasome degraded the substrate more slowly than Rpn13, Rpn10, and TM proteasome. Thus, the Rpn1 receptor was impairing degradation in this context (Fig. 8a, Supplementary Table 2). Rpn1 apparently positioned UBL-GFP-35 too far from the entrance of the degradation channel to allow the proteasome

to engage the protein[11] because increasing the length of the initiation region to 95 amino acids (UBL-GFP-95) led to efficient degradation (Fig. 8b, Supplementary Table 2). This degradation was inhibited by the purified UBL domain but not by K48- or K63-linked tetraubiquitin chains (Supplementary Fig. 9).

Rpn13 mediated degradation of proteins targeted to the proteasome by UBL domains at least fourfold faster than proteins targeted by single ubiquitin chains [initial rate of degradation of UBL-GFP-35 higher than of Ub$_5$(K48)-GFP-35 or Ub$_9$(K48)-GFP-35; Supplementary Table 2], suggesting that one of its key functions may be to mediate degradation through UBL-UBA proteins rather than recognizing ubiquitinated proteins directly. Similarly, Rpn1 mediated degradation of UBL-domain substrates roughly fourfold faster than proteins targeted by single ubiquitin chains [comparing the initial rate of degradation of UBL-GFP-95 to that of Ub$_5$(K48)-GFP-95; Supplementary Table 2], suggesting that Rpn1 may also play an important role in degradation mediated by UBL-UBA proteins.

Interestingly, UBL-GFP-95 was degraded well even by proteasome in which all three ubiquitin-binding sites were mutated (TM proteasome) (Fig. 8b, Supplementary Table 2). Deleting Rpn13 entirely did not inhibit UBL-GFP-95 degradation further (TMΔ13 proteasome), showing that degradation was not due to residual binding to the mutated binding site on Rpn13 (Fig. 8b, Supplementary Table 2). Mutating the binding site for the divergent UBL of Ubp6 that was recently identified on Rpn1 (the T2 site) (ref. [40]) does not inhibit degradation of UBL-GFP-95 (Supplementary Fig. 10). These observations suggest the existence of an as yet uncharacterized UBL receptor on the proteasome.

## Discussion

Ubiquitin can form chains of different lengths and linkages and modify proteins at one or more positions. The diversity of ubiquitin chain topologies allows them to mark proteins for a variety of fates. In this study, we investigated how the proteasome recognizes ubiquitinated proteins and find that the RP and the three Ub/UBL receptors identified to date provide a versatile recognition surface that can bind proteins modified by different ubiquitin chains in different configurations. Substrate binding and degradation are not strictly linked, and Ub/UBL receptors cooperate with one another in ways that depend on the arrangement of ubiquitin groups on the substrate.

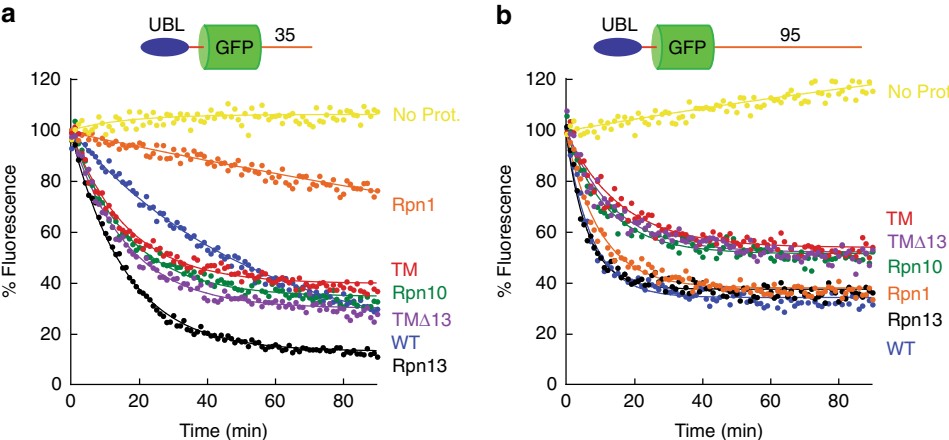

**Fig. 8 Degradation of UBL-targeted substrates.** Degradation of substrate proteins with ubiquitin-like domains derived from Rad23 (UBL) by the indicated proteasome types was followed under single-turnover conditions (5 nM substrate, 25 nM proteasome) in the presence of 1 mM ATP at 30 °C. The graphs show substrate fluorescence as a percentage of the initial fluorescence as a function of time in minutes. Proteasome types are described in Supplementary Table 1 (TM triple mutant proteasome). Each panel shows the degradation of particular substrates, and as follows: **a** UBL-GFP-35 and **b** UBL-GFP-95 on different proteasome mutants. Source data are provided as a Source Data file.

Ubiquitin chains linked through K48 and K11 are associated with proteasomal degradation physiologically, whereas chains linked through K63 or M1 (i.e., linked head to tail) are more commonly associated with other processes in the cell. We did not observe this pattern in our experiments with purified proteasome and substrates. Instead, the proteasome was able to degrade most but not all of the substrates presented to it. It preferred to degrade substrates with chains of four or more K48-linked ubiquitin molecules attached near the N terminus of the model substrates. However, the preference was mild and some substrates with a K63-linked chain or with two shorter K48-linked chains were degraded almost as well (at most 1.5-fold slower). Indeed, K48-linked ubiquitin chains were not the best degradation signals for all protein configurations. K63-linked chains were more effective degradation signals than K48 chains when the chains were attached near the C-termini of our substrates. The proteasome we used did not contain non-stoichiometric components, such as DUBs and substrate shuttles (UBL-UBA proteins), and it is under debate whether the presence of DUBs would attenuate degradation of substrates with multiple K63-linked chains[25,72]. In summary, the fact that proteins modified with K63-linked chains are rarely degraded by the proteasome physiologically is not due to a lack of recognition by the proteasome and has to be explained by another mechanism. For example, proteins with K63-linked chains and proteasome may simply be localized to different subcellular regions[26].

The proteasome did not degrade substrates with M1-linked chains (head to tail-linked ubiquitin molecules). These chains are synthesized physiologically by the E3 LUBAC and serve the formation of signaling complexes in the in the NFκB signaling pathway[64]. Thus, our finding here that they are not recognized by the proteasome fits well with their physiological role and explains the mechanism of specificity. We also found again, quite intriguingly, that the proteasome did not degrade proteins with single polyubiquitin chains linked through K11 (refs. [24,34]) (Fig. 4). K11-linked chains are associated with proteasome degradation in cell cycle regulation and several studies show that K11 chains can target substrates for proteasomal degradation. However, the K11 linkages typically occur in the context of branched ubiquitin chains containing mixed linkages in vitro and in vivo[35,36] so that substrates with a single homotypic ubiquitin chain may not be the correct model for these chains. The ends of branches of a mixed

ubiquitin chain may resemble the two ubiquitin domains in our multichain substrate.

We found that proteins modified with two ubiquitin chains are degraded particularly well and robustly even when the chains are short. Indeed, even substrates with two individual ubiquitin domains, mimicking proteins with multiple monoubiquitin tags, were degraded efficiently. The recognition of multichain substrates may be less dependent on ubiquitin linkage than that of substrates with single ubiquitin chains.

Increasing the length of K48-linked ubiquitin chains enhanced a substrate's affinity for the proteasome but did not lead to faster degradation. Unanchored chains of eight ubiquitin molecules bound the proteasome more tightly than chains of four molecules, and substrate with the longer chains bound the proteasome more tightly as reflected in lower $K_M$ values. Nevertheless, substrates with the longer and shorter chains were degraded equivalently in single-turnover experiments and had similar specificity constants ($k_{cat}/K_M$) in multiple turnover experiments. Thus, the proteasome did not select substrates with longer chains over those with shorter chains. This finding suggests that an increased substrate affinity can in fact slow protein turnover. One possible explanation is that longer ubiquitin chains lead the substrate to bind in configurations that make it more difficult for the proteasome to engage its substrate at the initiation site. Another possibility is that release of the ubiquitin chain becomes rate-limiting for the longer chains. In the cell, DUBs may trim the chains to their optimal length and UBL-UBA proteins may place larger substrates in productive configurations on the proteasome. In principle, longer chains may also enhance degradation by increasing the time required for a DUB to remove the chain and thus the time available for initiation by the proteasome. However, the most active DUBs on the proteasome, Ubp6/USP14 and Rpn11, both remove chains all together rather than through progressive chain trimming from the distal end of the chain[72,73].

The canonical view of substrate recognition, reaching back decades, is that while a ubiquitin chain length of four represents the minimal signal for targeting substrate to the proteasome, a chain of double that length has a higher proteasome affinity and is therefore a stronger targeting signal[5]. Our results may differ because the earlier study used a substrate [Ub$_4$(K48)-DHFR] that is degraded much more slowly than the substrates used in this study (the difference in $k_{cat}$ values to Ub$_5$(K48)-GFP-35 is

approximately 13-fold), presumably because the substrates in the earlier study contained no disordered regions that allow the proteasome to initiate degradation. Therefore, the contribution of any other steps to degradation of the DHFR substrates may be masked by the slow engagement and unfolding steps.

We find Rpn10 to be the primary receptor mediating the degradation of substrates targeted to the proteasome by ubiquitin chains. Rpn13 and Rpn1 by themselves presented these proteins for degradation only ineffectively. However, Rpn1 and to a lesser extent Rpn13 served as co-receptors with Rpn10 to enhance degradation of substrates with K63-linked chains and those with multiple chains. When the proteasome is saturated with substrate and has to turn over multiple times, Rpn13 may even slow the degradation of ubiquitinated substrates presented by Rpn10. Slower degradation could allow more time for deubiquitination and release of substrates, serving as a final checkpoint before a protein is degraded. Alternatively, substrate binding simultaneously to Rpn13 and a second place on the proteasome could constrain the conformational changes the proteasome undergoes between rounds of substrate degradation[1]. On mammalian proteasomes, Rpn13 binds the DUB Uch37 (refs. [74–76]), which is not present in *S. cerevisiae*. Ubiquitin chain editing by Uch37 could increase the stringency of substrate selection[77] by reducing the substrate affinity to, or residence time on, the proteasome. Rpn13 and Rpn1 proteasome degraded proteins targeted for degradation by a UBL domain as efficiently as the wild-type proteasome degraded proteins with the K48-linked ubiquitin chains. Thus, a key role for Rpn13 and Rpn1 may be to mediate protein degradation by recognizing substrates indirectly and serving as a receptor for UBL-UBA shuttle receptors[40,48] in addition to serving as co-receptors for substrates with multiple ubiquitin chains.

Rpn10 binds to ubiquitin chains through a small UIM domain, which is linked to the VWA domain of Rpn10 and thus the body of the proteasome through a flexible linker. This design makes the ubiquitin-binding domains mobile and allows them to take up a wide range of positions. Indeed, the UIM domains are not resolved in structures of the proteasome even when the main portion of Rpn10 (the VWA domain) is well resolved[48,78–82]. In contrast, Rpn13 binds ubiquitin through a compact PRU domain, and Rpn1 binds ubiquitin through its toroid domain, both of which are integral parts of the structure of the proteasomal RP and take up defined positions even if the subunits show some movement during the reaction cycle (e.g., ref. [83]). Thus, Rpn10, in contrast to Rpn13 or Rpn1, is likely to be able to position substrate proteins bound through their ubiquitin chains in many orientations, allowing the receptor to place proteins of different structures in ways that the proteasome can engage them at an unstructured region. This flexibility may explain why Rpn10 is able to function as the primary receptor for substrates targeted for degradation by polyubiquitin chains. The Ub/UBL-binding site of Rpn10 is likely closer to the pore-1 loops in the Rpt ring than the binding sites of Rpn1 and Rpn13 (Fig. 1) and this proximity may also facilitate substrate presentation by Rpn10. In UBL-UBA proteins, the proteasome- and ubiquitin-binding domains are connected by flexible linkers[84], which may reintroduce the plasticity in substrate placement in addition to considerably closing the distance between the receptor and the translocation channel.

Interestingly, no individual ubiquitin receptor appeared to be strictly required for degradation of multichain conjugates, as mutating any two receptors affected degradation only mildly (Fig. 7a). Even substrates with two individual ubiquitin domains, mimicking proteins with multiple monoubiquitin tags, were degraded efficiently. Similarly, substrate proteins in which four ubiquitin molecules were linked to each other in frame by short linkers (Ub$_4$lin) degraded robustly, perhaps mimicking a protein with four monoubiquitin modifications. It is often difficult to

prevent ubiquitination of a protein by mutating individual lysine residues or even combinations of lysines, which suggests that ubiquitination can occur on several residues within one protein and that multichain conjugates may be common in cells. Ubiquitination simultaneously at several locations in the same protein enhanced degradation of at least some cell cycle regulators such as cyclin B[69] and Sic1 (ref. [85]). Similarly, branched ubiquitin chains enhance degradation of other cell cycle regulators and structurally resemble multiple ubiquitin chains after the branch point[16,35,36,63].

Genetic and biochemical experiments suggest that not all ubiquitin receptors have been identified[40], and we see evidence for cryptic substrate receptors here. Proteins with single K48-linked ubiquitin chains were degraded slowly but substantially by proteasome in which the Rpn1, Rpn10, and Rpn13 ubiquitin-binding sites had been mutated simultaneously. Degradation of the multichain substrate on TM proteasome was even more striking, as mutating all three ubiquitin receptors reduced proteolysis only fivefold. Some residual binding of ubiquitinated substrates has been observed on Rpn1-ARR[40], and thus the degradation of protein with a K48-linked ubiquitin chain on TM proteasome could be due to residual binding to this site. However, this mechanism seems unlikely because restoring the Rpn1 receptor (Rpn1 proteasome) did not improve degradation of the Ub$_4$(K48) substrate over TM proteasome (Fig. 2a, Supplementary Table 2). Thus, the residual degradation is presumably mediated by binding to an alternate location, perhaps a ubiquitin-binding site that has not yet been identified. The central section of proteasome subunits Rpn1 and Rpn2 consist of a repetitive structure built from a spiral of α-helices packing against each other to form a toroid. In Rpn1, the two grooves between neighboring α-helices form binding sites for ubiquitin or ubiquitin-like domains[40]. It is possible that the grooves between other helices in these proteins form additional weaker binding sites that become relevant when ubiquitin chains present on proteins are placed in their vicinity. In any case, the proteasome provides a surface that allows multiple similarly effective binding modes.

Experiments with substrates targeted to degradation by UBL domains also indicate the presence of an as yet undefined substrate receptor. Mutating Rpn10, Rpn13, and Rpn1 simultaneously reduced degradation of UBL-GFP-95 only approximately threefold. The remaining degradation was not due to residual binding to Rpn13 because deleting Rpn13 entirely did not affect degradation further. No residual binding of the UBL domain was observed on Rpn1-ARR[40]. The proteasome used here did not contain Ubp6; however, mutating the recently identified binding site for Ubp6's UBL domain[40] in addition to the established ubiquitin/UBL receptors did not affect degradation.

In summary, we describe how the three established ubiquitin receptors on the proteasome, Rpn10, Rpn13, and Rpn1, recognize proteasome substrates with defined ubiquitin chains. We find that substrate binding to receptors does not necessarily lead to degradation, and some receptors can even inhibit degradation of specific substrates effectively targeted for degradation by another. Rpn10 is the primary receptor that targets ubiquitinated proteins for degradation, whereas Rpn1 serves as a co-receptor for substrates with multiple ubiquitin chains. Rpn13 and Rpn1 were the primary receptors for substrates targeted for degradation by UBL domains and may thus serve as receptors for substrates delivered by UBL-UBA shuttle receptors. Proteins modified with multiple ubiquitin chains are degraded robustly even if these chains are short. Degradation of multichain substrates is barely affected by attenuation of any individual ubiquitin receptor, suggesting that multiple modifications on single proteins serve as powerful signals to clear proteins from cells, mediated by a useful synergy among the three receptors. The RP of the proteasome provides a

versatile recognition surface that can recognize and degrade substrates with a multitude of different configurations of ubiquitin chains and initiation region. The versatility of the binding platform is provided by the multiplicity of the Ub/UBL receptors, their broad, flexible distribution over the surface of the RP, and the differences in binding specificities of the receptors.

## Methods

**Molecular biology**. Ub-GFP-35, 35-GFP-Ub, and Ub-35-Ub-GFP-35 were described previously[24] and represent fusions of the coding sequences for ubiquitin (Ub) or the first 35 amino acids of cytochrome $b_2$ (35), both from S. cerevisiae, or for the circular permutant CP8 of superfolder GFP from Aequorea victoria[60]. Ub-GFP-95 and 95-GFP-Ub were derived from Ub-GFP-35 and 35-GFP-Ub, respectively, by replacing the 35 residue region with the sequence encoding the first 95 amino acids S. cerevisiae cytochrome $b_2$ (RLRYQPLLRI SQNCEAAILR ASQTRL NTIG AYGSTVPRSQ SFEQDSRQRT QSWTALRVGA IPAATSSVAY LNWH NGQIDN EPQLDMNRQR ISPAE)[9]. UBL-GFP-35 and UBL-GFP-95 were derived from Ub-GFP-35 and Ub-GFP-95, respectively, by replacing the Ub coding region with that for the first 80 amino acids of S. cerevisiae Rad23. Ub$_5$(M1)-GFP-35 was derived from Ub-GFP-35 by replacing the Ub coding region with that of the UBI4 gene from S. cerevisiae genomic DNA. DNA sequences of all constructs are shown as Supplementary Note 2 in the Supplementary Information. The sequences of the primers used are given in Supplementary Table 5.

**Protein expression and purification**. Ub and Ub$_4$(M1), and Ub(K48R) were purified using established protocols[24,86,87]. Proteins were expressed in E. coli Rosetta (DE3) pLysS (Novagen) cells from pET3a-derived plasmids through induction with 0.4 mM dioxane-free isopropyl β-D-1-thiogalactopyranoside (IPTG) (Calbiochem) at an OD$_{600}$ of 0.6 for 4 h at 37 °C. Cells were harvested by centrifugation at 5000 × g for 10 min and resuspended in 50 mM Tris-HCl pH 7.6 and stored at −80 °C. For protein purification, cells were thawed in a room-temperature water bath in the presence of Protease Inhibitor Cocktail Set V, EDTA-Free (Calbiochem catalog no. 539137), lysed by two passages at 15,000 psi through a high-pressure homogenizer (EmulsiFlex-C3, Avestin), and the lysate cleared by centrifugation (8000 × g for 20 min at 4 °C). Perchloric acid was added to clarified lysate (0.5% (v/v)) and precipitated proteins were removed by centrifugation (8000 × g for 20 min at 4 °C). The supernatant was dialyzed against 50 mM ammonium acetate pH 4.5, passed through a 0.45 μm syringe filter and protein was purified by cation exchange chromatography in 50 mM ammonium acetate pH 4.5 by applying a gradient of 0 to 500 mM NaCl to a Resource S 6 mL (GE) column. Finally, purified protein was concentrated to 50−100 mg mL$^{-1}$ and buffer exchanged into 50 mM Tris-HCl pH 7.6 using an Amicon Ultra Centrifugation filter with a 3 kDa molecular weight cutoff.

His-HRV3C-Ub(K48R), His-HRV3C-Ub(K63R), His-HRV3C-Ub(K11R), Ub$_4$(lin)-His, UBL-His, Ub-GFP-35-His, His-35-GFP-Ub, Ub-35-Ub-GFP-35-His, Ub-GFP-9-His, His-95-GFP-Ub, Ub$_4$(lin)-GFP-35-His, UBL-GFP-35-His, and UBL-GFP-95-His were expressed as fusion proteins containing a 6xHis-tag and purified by nickel-affinity chromatography following published protocols[24]. Proteins were expressed in E. coli BL21 (DE3) cell from pGEM- or pET Duet-derived plasmids through induction with 0.4 mM dioxane-free IPTG at an OD$_{600}$ of 0.6 for 4 h at 37 °C. Cells were harvested by centrifugation at 5000 × g for 10 min, resuspended in NPI-10 buffer (10 mM imidazole, 300 mM sodium chloride, 50 mM sodium phosphate buffer pH 8) and stored at −80 °C. Cells were thawed in a room-temperature water bath after addition of Protease Inhibitor Cocktail Set V, EDTA-Free (Calbiochem catalog no. 539137), DNase I, and 10 mM MgCl$_2$, lysed in a high-pressure homogenizer as described above and the lysate cleared by centrifugation at 30,000 × g for 20 min at 4 °C. The clarified lysate was passed through a through a 0.45 μm syringe filter and applied to a pre-washed nickel sepharose column (5 mL HisTrap FF Crude (GE)) on an FPLC and washed first with two column volumes (CV) of NPI-10 and then with 10 CV of NPI-20 (20 mM imidazole, 300 mM sodium chloride, 50 mM phosphate buffer pH 8.0). Proteins were eluted with 10 CV of NPI-250 (250 mM imidazole, 300 mM sodium chloride, 50 mM phosphate buffer pH 8.0). The ubiquitin variants were then concentrated and buffer exchanged into 50 mM Tris-HCl pH 7.6 at a final concentration of 20−50 mg mL$^{-1}$ using an Amicon Ultra Centrifugation filter with a 3 kDa molecular weight cutoff whereas the base proteins were concentrated and buffer exchanged into 50 mM Tris-HCl pH 7.6 and 5% glycerol using an Amicon Ultra Centrifugation filter with a 10 kDa molecular weight cutoff. Ub$_5$(M1)-GFP-35 was isolated by affinity chromatography in two steps following published protocols[24,87]. The protein was first enriched through its C-terminal 6xHis-tag as described for the base proteins above and then further purified through its N-terminal GST-tag. The nickel sepharose eluate was concentrated and buffer exchanged into 50 mM Tris-HCl pH 7.6 using an Amicon Ultra Centrifugation filter with a 10 kDa molecular weight cutoff. It was then mixed with 2 mL of washed Glutahione Sepharose 4B beads (GE), allowed to bind under nutation at 4 °C for 1 h, and collected in a PD-10 column. The column was washed first with 10 CV GST wash buffer (1× PBS, 1 mM DTT) and then with 10 CV GST PreScission cleavage buffer (150 mM sodium chloride, 1 mM EDTA, 1 mM DTT, 50 mM Tris-HCl pH 7.4). Finally, the protein was incubated with PreScission protease in GST PreScission

cleavage buffer overnight at 4 °C and eluted with 3 CV of cleavage buffer. The eluate was concentrated and buffer exchanged into 50 mM Tris-HCl pH 7.6 and 5% glycerol using an Amicon Ultra Centrifugation filter with a 10 kDa molecular weight cutoff.

E2-25K (Homo sapiens), Ubc13 (S. cerevisiae), Mms2 (S. cerevisiae), and UBE2S-UBD were purified as GST-fusion proteins and the GST-tag was removed by PreScission Protease according to the published prototcol[87] with some modifications as described. The UBE2S-UBD expression plasmid was a gift from David Komander. Proteins were expressed in E. coli Rosetta (DE3) pLysS (Novagen) cells from pGEX-6P-1-derived plasmids through induction with 0.4 mM dioxane-free IPTG at an OD$_{600}$ of 0.6 for 4 h at 37 °C. Cells were harvested by centrifugation at 5000 × g for 10 min, resuspended in 1× PBS and stored at −80 °C. Cells were thawed in a room-temperature water bath after addition of 1% TritonX-100, 1 mM DTT, Protease Inhibitor Cocktail Set V, EDTA-Free (Calbiochem catalog no. 539137), DNase I, and 10 mM MgCl$_2$. Cells were then lysed in a high-pressure homogenizer as described above and the lysate nutated for 30 min at 4 °C to allow the fusion proteins to solubilize. Lysate was then cleared by centrifugation at 30,000 × g for 20 min at 4 °C. The clarified lysate was passed through a 0.45 μm syringe filter and applied to washed Glutathione Sepharose 4B beads (GE). Proteins were purified through their GST-tag and the tag removed as described above for Ub$_5$(M1)-GFP-35.

The Ube1 (Mus musculus) expression plasmid was a gift from Jorge Eduardo Azevedo (Addgene plasmid # 32534) and the protein was purified through an N-terminal 6xHis-tag following published protocols[88] with some modification as described. The protein was expressed in E. coli BL21(DE3) cells from a pET28-derived plasmid through induction with 0.5 mM dioxane-free IPTG (Calbiochem) at on OD$_{600}$ of 0.6 overnight at 16 °C. Cells were harvested by centrifugation at 5000 × g for 10 min, resuspended in 50 mM Tris-HCl pH 8.0, and stored at −80 °C. Cells were thawed in the presence of 150 mM NaCl, 0.1% TritonX-100, 1 mM EDTA, 1 mM DTT, and Protease Inhibitor Cocktail Set V, EDTA-Free (Calbiochem catalog no. 539137) in a water bath at room temperature, lysed with a high-pressure homogenizer as described above, and the lysate clarified by centrifugation at 30,000 × g for 20 min at 4 °C. The clarified lysate was passed through a 0.45 μm syringe filter, mixed with 1.5 mL of washed Ni-NTA beads (Qiagen catalog no. 30210), and allowed to bind under nutation at 4 °C for 1 h. The mixture was then poured into an PD-10 column, allowed to settle by gravity, and washed three times with 10 CV of NPI-10. Protein was eluted with three washes of 2 CV of NPI-250, and the eluate concentrated to 1 mL and buffer exchanged into 10 mM Tris-HCl pH 8, 1 mM EDTA, and 1 mM DTT using an Amicon Ultra Centrifugation filter with a 30 kDa molecular weight cutoff.

The AMSH* (Mus musculus) expression plasmid was a gift from David Komander. The protein was expressed as a fusion with an N-terminal 6xHis-tag and an HRV3C cleavage site, concentrated by affinity chromatography, the 6xHis-tag removed by HRV3C protease cleavage, and finally purified by size-exclusion chromatography following published protocols[89–91] with some modifications as described. The protein was expressed in E. coli Rosetta (DE3) pLysS cells (Novagen) from a pOPINB-derived plasmid through induction with 0.2 mM dioxane-free IPTG at an OD$_{600}$ of 0.8 for 16 h at 18 °C. Cells were harvested by centrifugation at 5000 × g for 10 min, resuspended in NPI-10 buffer, and stored at −80 °C. Cells were thawed in the presence of 1 mM DTT, Protease Inhibitor Cocktail Set V, EDTA-Free (Calbiochem catalog no. 539137), DNase I, and 10 mM MgCl$_2$ in a room-temperature water bath, lysed in a high-pressure homogenizer as described above. The lysate was cleared by centrifuging twice at 15,000 × g for 20 min at 4 °C, passed through a 0.45 μm syringe filter, and mixed with 2 mL of washed Ni-NTA beads (Qiagen catalog no. 30210). The mixture was incubated under nutation at 4 °C for 1 h, poured into an empty PD-10 column, and allowed to settle by gravity. The column was washed twice with 10 CV of NPI-10 supplemented with 1 mM DTT followed by a wash with 10 CV of HRV3C cleavage buffer (50 mM Tris-HCl pH 7.4, 150 mM NaCl, 1 mM DTT). His-HRV3C protease in HRV3C cleavage buffer was then added to the column, the column capped, and nutated overnight at 4 °C. Protein was eluted with 3 CV of HRV3C cleavage buffer and concentrated to 0.5 mL using an Amicon Ultra Centrifugation filter with a 10 kDa molecular weight cutoff. Finally, the concentrated eluate was separated on a Superdex Hi-Load 75-pg (GE catalog no. 28-9893-33) column in AMSH* size-exclusion buffer (50 mM Tris-HCl pH 7.6, 150 mM NaCl, 4 mM DTT) at a flow rate of 0.25 mL min$^{-1}$, collecting 2 mL fractions. Fractions containing protein were concentrated and buffer exchanged into 50 mM Tris-HCl pH 7.6 using an Amicon Ultra Centrifugation filter with a 10 kDa molecular weight cutoff.

**Generating ubiquitin chains**. Ub$_4$(K48), Ub$_4$(K63), Ub$_8$(K48), and Ub$_4$(K11) ubiquitin chains were generated using enzymes that are part of the natural synthesis machinery and purified following published protocols[24,87] as described below. Chains were synthesized from a mixture of wild-type ubiquitin and an N-terminally 6xHis-tagged ubiquitin in which the relevant lysine residue was mutated to arginine so chain synthesis terminated with the His-tagged ubiquitin molecule. Polyubiquitin chains of different lengths were then isolated in three steps: an initial enrichment using the N-terminal 6xHis-tag (elution with HRV3C protease) was followed by purification on a cation exchange column (Resource S) using a linear salt gradient and finally size-exclusion chromatography[24,87] (Superdex 75).

K48-linked ubiquitin chains were generated using *H. sapiens* ubiquitin-conjugating enzyme E2-25K[92] together with the *M. musculus* ubiquitin activating enzyme (E1) Ube1 (ref. [93]) acting on a mixture of wild-type *S. cerevisiae* ubiquitin and a mutant of *S. cerevisiae* ubiquitin carrying the mutation K48R and an N-terminal 6xHis-tag attached through a linker containing an HRV3C protease cleavage site (His-HRV3C-Ub(K48R)). Chains of two or four ubiquitin moieties in length were synthesized by incubating 7.5 mg mL$^{-1}$ wild-type ubiquitin and 7.5 mg mL$^{-1}$ His-HRV3C-Ub(K48R), in one-fifth volume of PBDM8 buffer (250 mM Tris-HCl pH 8.0, 25 mM MgCl$_2$, 50 mM creatine phosphate, 3 units mL$^{-1}$ inorganic pyrophosphatase, 3 units mL$^{-1}$ creatine phosphokinase), 2.5 mM ATP, and 0.5 mM DTT, with 20 μM E2-25K, and 0.2 μM E1 at 37 °C overnight. Chains of eight ubiquitin moieties were synthesized using the same protocol but incubating 11.25 mg mL$^{-1}$ wild-type ubiquitin and 3.75 mg mL$^{-1}$ His-HRV3CUb(K48R). The reactions were quenched with 5 mM DTT and aggregates were removed by centrifugation at 15,000 × *g* for 5 min at 4 °C. The supernatants were then diluted with an equal volume of NPI-10, 1 mL of washed Ni-NTA beads (Qiagen catalog no. 30210) were added for each 50 mg of total ubiquitin in the reaction, and the mixtures were allowed to bind under nutation at 4 °C for 1 h. The mixture was then poured into an empty PD-10 column and allowed to settle by gravity. The column was washed twice with 10 CV of NPI-10, followed by 10 CV of HRV3C cleavage buffer (50 mM Tris-HCl pH 7.4, 150 mM NaCl, 1 mM DTT). His-HRV3C protease in HRV3C cleavage buffer were then added to the column, the column capped and nutated overnight at 4 °C. Finally, the chains were eluted with 1 CV HRV3C cleavage buffer. For the next purification step by cation exchange chromatography, the eluate was acidified by the addition of 0.03 volumes of 2 N acetic acid to a pH of 4 and loaded on to 6 mL Resource S (GE catalog no. 17-1180-01) in a Tricorn column equilibrated with 50 mM ammonium acetate pH 4.5. The column was washed with 2 CV of 50 mM ammonium acetate pH 4.5 and the chains were eluted in the same buffer with a NaCl gradient as follows: 1 CV of 0–200 mM NaCl, 25 CV of 200–450 mM NaCl, and 1 CV of 450–1000 mM NaCl in 50 mM ammonium acetate pH 4.5, taking 2 mL fractions. The peak containing the chains of the desired length was concentrated to 0.5 mL using an Amicon Ultra Centrifugation filter with a 3 kDa molecular weight cutoff. The chains were purified further on a Superdex Hi-Load 75-pg (GE catalog no. 28-9893-33) column at a rate of 0.25 mL min$^{-1}$ in ubiquitin size-exclusion buffer (150 mM NaCl, 50 mM Tris-HCl pH 7.6, 0.5 mM EDTA), collecting 2 mL fractions. Fractions containing chains of the desired length were concentrated and exchanged into 50 mM Tris-HCl pH 7.6 using an Amicon Ultra Centrifugation filter with a 3 kDa molecular weight cutoff.

K63-linked chains were synthesized using *S. cerevisiae* E1 and the E2/E2 variant complex Ubc13/Mms2 (refs. [22,94]). Wild-type ubiquitin at 10 mg mL$^{-1}$ (to make chains of four ubiquitin moieties) or 20 mg mL$^{-1}$ (to make chains of eight ubiquitin moieties) and His-HRV3C-Ub(K63R) ubiquitin at 5 mg mL$^{-1}$ were incubated with one-fifth volume of PBDM7.6 buffer (250 mM Tris-HCl pH 7.6, 25 mM MgCl$_2$, 50 mM creatine phosphate, 3 units mL$^{-1}$ inorganic pyrophosphatase, 3 units mL$^{-1}$ creatine phosphokinase, and 10 mM ATP), 0.5 mM DTT, 8 μM Ubc13, 8 μM Mms2, and 0.2 μM E1 at 37 °C overnight. The chains were then purified as described above for K48 chains.

K11-linked ubiquitin chains were generated by the same method as K63-linked chains except that N-terminally His-tagged K11R ubiquitin (His-HRV3C-Ub (K11R)) was used with 80 μM E2 UBE2S-UBD, 0.3 μM E1, and 0.4 μM AMSH* (ref. [90]). Chains were again isolated by nickel-affinity, ion-exchange, and size-exclusion chromatography[24,87] as described above for K48 and K63 chains.

**Generation of ubiquitinated substrate**. Ubiquitinated substrates were generated following published protocols[24]. Polyubiquitin chains were attached to the ubiquitin domain or domains in the target protein using the same enzymes used to create the polyubiquitin chains. However, the Ubc13/Mms2 complex will only attach ubiquitin efficiently to a ubiquitin with a free N terminus and therefore cannot be used to attach chains to C-terminal or internal ubiquitin domains. Therefore, K63-linked chains had to be attached to the C-terminal ubiquitin domains using E2-25K, which can attach ubiquitin chains to ubiquitin domains in which either or both termini are blocked. K63-linked ubiquitin chains for attachment to the C-terminal end of substrate proteins were generated by the Ubc13/Mms2 complex but using the Ub(K48R) variant instead of wild-type ubiquitin together with His-HRV3C-Ub(K63R). The chains were then attached to a C-terminal ubiquitin domain within the target protein using E2-25K and thus creating a substrate with one proximal K48 linkage and K63 linkages throughout the rest of the chain. After incubation with enzymes and free ubiquitin chains, the reaction was quenched with 5 mM DTT and aggregates were removed by centrifugation at 15,000 × *g* for 5 min at 4 °C. Base protein was separated from unreacted ubiquitin chains and enzymes by nickel-affinity chromatography using the C-terminal His-tag on the base protein. The reaction mixture was diluted with an equal volume of NPI-10, and the protein applied to a pre-washed nickel sepharose column (5 mL HisTrap FF Crude (GE)). The column was washed with 2 CV of NPI-10 and 10 CV of NPI-20, and eluted with 10 CV of NPI-250. Finally, modified and unmodified base protein were separated by size-exclusion chromatography. The eluate from the nickel column was concentrated to 0.5 mL using an Amicon Ultra Centrifugation filter with a 30 kDa molecular weight cutoff and applied to a Superdex Hi-Load 200-pg (GE catalog no. 28-9893-35) column at a

flow rate of 0.25 mL min$^{-1}$ in ubiquitin size-exclusion buffer supplemented with 1 mM DTT, collecting 2 mL fractions. Fractions containing ubiquitinated substrate were pooled, concentrated, and buffer exchanged into 50 mM Tris-HCl pH 7.6 with 5% glycerol using an Amicon Ultra Centrifugation filter with a 30 kDa molecular weight cutoff.

K48-linked ubiquitin chains were attached to base proteins by incubating 15 μM ubiquitin chain with 45 μM Ub-GFP-tail or tail-GFP-Ub, or 45 μM chain with 15 μM Ub-35-Ub-GFP-tail, using the same enzymes and conditions as for K48-linked ubiquitin chain synthesis. K63-linked chains were attached to base protein by incubating 15 μM chain with 45 μM base protein using the same conditions as for K63-linked ubiquitin chain synthesis when the chains were attached at an N-terminal ubiquitin. K63-linked chains were attached base proteins at the C-terminal ubiquitin using the same enzymes and buffer conditions as for K48-linked ubiquitin chain synthesis. K11-linked chains were attached to base proteins by incubating 15 μM ubiquitin chain with 45 μM Ub-GFP-tail using the same buffer conditions but with 60 μM E2 UBE2S-UBD, 0.25 μM E1, and 0.4 μM AMSH*. Ubiquitin chain and base protein attachment reactions were incubated at 37 °C overnight.

**Proteasome purification**. Proteasome was purified from *S. cerevisiae* following published protocols[95] with minor modifications described below. The CP and RP of the proteasome were purified separately using FLAG-tags fused to Pre1 and Rpn11, respectively. The CP was purified from yeast strains YYS37 (ref. [95]) and the RPs were purified from yeast strains as indicated (Supplementary Table 1). Strains generated for this study were constructed using standard methods as described[40] and are isogenic to SUB61 (*MAT α lys2-801 leu2-3,2-112 ura3-52 his3-Δ200 trp1-1(am)*)[96].

Starter cultures of 5 mL YPD with 2% glucose were inoculated with single colonies and grown for 24 h at 30 °C. The culture was diluted 1:100 into 2 L of YPD with 2% glucose and grown at 30 °C to OD$_{600}$ ~2. Cells were harvested by centrifugation at 5000 × *g* for 10 min, washed with ice-cold water, and then with buffer A (50 mM Tris-HCl pH 7.4, 10% glycerol), re-pelleted at 5000 × *g* for 10 min and stored at −80 °C. For purification, cell pellets were resuspended in buffer C (50 mM Tris-HCl pH 7.4, 10% glycerol, 1 mM ATP, 10 mM MgCl$_2$, 1 mM DTT, 20 mM creatine phosphate, and 0.02 mg mL$^{-1}$ creatine phosphokinase) and lysed by two passages at ~30,000 psi in a high-pressure homogenizer (EmulsiFex-C3, Avestin). Lysates were clarified by centrifugation at 30,000 × *g* for 30 min at 4 °C, passed through a 0.45 μm syringe filter, and supplemented with 5 mM ATP, 0.01 mg mL$^{-1}$ creatine phosphokinase, and 10 mM creatine phosphate. Clarified lysate was mixed with 1 mL pre-washed anti-FLAG M2-agarose beads (Sigma) and allowed to bind under nutation at 4 °C for 2 h. The mixture was collected in a PD-10 column, allowed to settle by gravity, washed twice with 15 CV of buffer B (50 mM Tris-HCl pH 7.4, 10% glycerol, 2 mM ATP, 5 mM MgCl$_2$, 1 mM DTT), then washed twice with 15 CV of buffer B supplemented with 500 mM NaCl. The column was filled with buffer B supplemented with 500 mM NaCl, capped, and nutated for 1 h at 4 °C. The mixture was allowed to settle by gravity. The column was washed twice with 15 CV of buffer B supplemented with 500 mM NaCl, washed twice with 15 CV of buffer B, and excess buffer removed by centrifugation at 500 × *g* for 5 s in a swinging bucket rotor. Proteasome was then eluted once with 0.75 mL of 5 mg mL$^{-1}$ 3xFLAG peptide in buffer B, then again with 0.5 mL of 0.15 mg mL$^{-1}$ FLAG peptide in buffer B. Eluate was pooled, aliquoted, flash frozen with liquid nitrogen, and stored at −80 °C.

For routine degradation assays, proteasomes were reconstituted by incubating CP with RP at a molar ratio of 1:2 for 30 min at 30 °C. Proteasome concentrations reported are based on the concentration of CP present in assembled proteasomes (i.e., 25 nM proteasome contains 25 nM CP and 50 nM RP).

Mammalian proteasome was purified from *Psmd14*$^{Flag/Flag}$ mouse embryonic fibroblasts (Rpn11-Flag MEFs) by anti-Flag immunoaffinity chromatography[65]. Rpn11-Flag MEFs were lysed with ice-cold purification buffer (20 mM Tris-HCl pH 7.4, 0.2% (v/v) NP-40, 150 mM NaCl, 1 mM DTT, 2 mM ATP, and 5 mM MgCl$_2$), clarified by centrifugation (20,000 × *g* for 10 min at 4 °C), immunoprecipitated using anti-Flag M2-agarose affinity beads (Sigma), and eluted with 0.15 mg mL$^{-1}$ Flag peptide (Sigma) in purification buffer.

**Kinetic plate reader assays**. Single-turnover degradation assays were performed as previously described[24]. The substrate proteins were presented to purified yeast proteasome and their degradation followed measuring GFP fluorescence. Typically, 25 nM reconstituted proteasome was incubated with 5 nM substrate in the presence of 1 mM ATP and an ATP-regenerating system (ARS) at 30 °C in 384-well plates (flat bottom, low flange, non-binding black plates; Corning #3575). Proteasome was reconstituted at twice the final concentration by incubating 50 nM CP with 100 nM RP in the presence of 8 mM DTT, degradation buffer (10 mM Tris-HCl pH 7.6, 1 mM MgCl$_2$, 1% glycerol), and 2× ARS (2 mM ATP, 20 mM creatine phosphate, and 0.2 mg mL$^{-1}$ creatine phosphokinase) at 30 °C for 30 min. Substrate was diluted to 10 nM in protein buffer (4 mg mL$^{-1}$ BSA in degradation buffer). Reactions were initiated by mixing 20 μL proteasome mixture and 20 μL substrate. Thus, the final reaction contained 5 nM substrate and 25 nM proteasome in 1 mM ATP, 10 mM creatine phosphate, 0.1 mg mL$^{-1}$ creatine phosphokinase, 4 mM DTT, degradation buffer, and 2 mg mL$^{-1}$ BSA.

The substrates' fluorescence (488 nm excitation; 520 nm emission) was measured every minute for 90 min in a plate reader (Infinite M1000 PRO, Tecan). Background (average fluorescence of a well containing everything except substrate) was subtracted and fluorescence plotted as a function of time. The decay curves were fitted to the equation describing a single exponential decay to a constant offset using the software package KaleidaGraph (version 4.1, Synergy Software). Initial rate was determined by multiplying the amplitude of degradation by the rate constant and converted to nM min$^{-1}$ assuming 100% of the fluorescence corresponded to 5 nM substrate and 0% of the fluorescence corresponded to 0 nM substrate.

**Inhibition assays**. Reconstituted proteasome was mixed to a concentration of 50 nM CP and 100 nM RP with 8 mM DTT in degradation buffer and two times ARS as above; substrate was diluted to 20 nM in protein buffer (4 mg mL$^{-1}$ BSA in degradation buffer). The inhibitor was diluted serially in degradation buffer at four times the desired final inhibitor concentration. In all, 10.5 µL of 20 nM substrate and 10.5 µL of the inhibitor cocktail were mixed. Proteasome and substrate–inhibitor mixtures were each incubated at 30 °C for 5 min and then 20 µL of substrate–inhibitor mixture (final substrate concentration 5 nM) was mixed with 20 µL of proteasome mix (final proteasome concentration 25 nM) in a 384-well plates. The substrate fluorescence (488 nm excitation, 520 nm emission) was measured every minute for 90 min in a plate reader (Infinite M1000 PRO, Tecan). Background (average fluorescence of a well containing everything except substrate) was subtracted and fluorescence plotted over time. The decay curves were fitted to the equation describing a single exponential decay to a constant offset using the software package KaleidaGraph (version 4.1, Synergy Software). The initial rate of degradation was calculated by multiplying the rate constant by the amplitude and plotted as a function of concentration, and the resulting curve was fit by $y/(1 + [I]/K_i)$ where $y = V_{max}[S]/K_M$ to determine $K_i$.

**Michaelis–Menten assays**. Reconstituted proteasome was mixed to a concentration of 50 nM CP and 100 nM RP with 8 mM DTT in degradation buffer and two times ARS. Substrate was diluted serially to twice the desired final concentration in protein buffer (4 mg mL$^{-1}$ BSA in degradation buffer). Each mixture was incubated at 30 °C for 5 min and then 20 µL of substrate was mixed with 20 µL of proteasome mix (final proteasome concentration of 25 nM) in a 384-well plate. Substrate fluorescence (488 nm excitation 520 nm emission) was measured every minute for 90 min in a plate reader (Infinite M1000 PRO; Tecan). Background (average fluorescence of a well containing everything except substrate) was subtracted and fluorescence plotted as a function of time. Initial rate was calculated by fitting a line through the initial linear portion of the curve. To convert initial rate from a.u. min$^{-1}$ to nM min$^{-1}$ the initial rate was divided by a correction factor for the a.u. nM$^{-1}$ of the specific substrate at the specific plate reader settings. The initial rate was plotted as a function of substrate concentration and fit to a Michaelis–Menten curve to determine $K_M$.

**Native proteasome gels**. Proteasome reconstitutions were resolved on 3.5% native polyacrylamide gels and then imaged by suc-LLVY-AMC cleavage activity[97] or through staining with Instant Blue. For the activity assays, gels were incubated 50 µM suc-LLVY-AMC in 50 mM Tris-HCl pH 7.4, 5 mM MgCl$_2$, and 1 mM ATP without agitation for 10 to 30 min at 30 °C. suc-LLVY-AMC is cleaved by the chymotryptic site in the CP and the proteolysis product can be visualized with a UV transilluminator[97].

**Reporting summary**. Further information on research design is available in the Nature Research Reporting Summary linked to this article.

## Data availability
The source data underlying Figs. 2–8 and Supplementary Figs. 2B, 3, 4, 7, 8, 9, 10 are provided as a Source Data file. The data that support the findings of this study are available from the corresponding authors upon reasonable request.

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

## Acknowledgements

These studies were supported by National Institutes of Health grants R01GM12450 and R01GM043601 as well as Welch Foundation grant F-1817.

## Author contributions

K.M.-F, C.D., T.T. and S.E. designed, conducted experiments and interpreted data; A.R.N. and Y.S. constructed unique reagents for the studies, and D.F. and A.M. conceived the study, designed experiments, and interpreted data. K.M.-F., C.D., T.T., S.E., A.R.N., Y.S., D.F. and A.M. wrote the manuscript.

## Competing interests

A.M. is a paid consultant of Kymera Therapeutics. The other authors declare no competing interests.
