## [Peer Review File · Nature Communications]

Reviewers' comments:

Reviewer #1 (Remarks to the Author):

In the current manuscript entitled "The Proteasome 19S Cap and its Three Identified Ubiquitin Receptors Provide a Versatile Recognition Platform for Substrates", Matouschek and colleagues define the molecular mechanism of how proteasomal ubiquitin receptors discriminate ubiquitin chains of different topologies using fully reconstituted system. In this well designed and carefully prepared manuscript, the authors clearly demonstrate that Rpn10 is the primary Ub receptor of the proteasome and that Rpn13 has an inhibitory role for proteasomal degradation. Moreover, the authors redefine the role of Rpn1 and Rpn13 as UBL receptors. Overall, I found this manuscript to be a fascinating read that describes very important research in future proteasome studies. However, a few issues should be addressed to strengthen the several of the conclusions.

Major points:

1. Throughout the study, proteasomal degradation was only analyzed by monitoring fluorescence loss of GFP. Are the substrate proteins truly degraded? Also, Ub5(K11) substrate and Ub5(M1) substrate were not degraded at all, but considering that these type of ubiquitin chains can nonselectively bind to the proteasome, they may be deubiquitinated. It is necessary to confirm the proteasomal degradation/deubiquitination by western blot analysis at least some timepoints of the unfolding assay.
2. There is no data for degradation assay of monomeric Ub-fused GFP-tail. The authors only presented the data of GFP-35 in Figure S3. Is Ub-GFP-tail not degraded at all? If Ub-GFP-tail is degraded by the proteasome, K11 and M1-linked ubiquitin chains have an inhibiting role for degradation.
3. The authors discuss that Rpn10 is the main Ub receptor and that Rpn1 and Rpn13 mainly function as UBL receptors. So, it is assumed that Rad23 binds to Rpn1 or Rpn13 and passes the Rad23-bound K48Ub substrate to Rpn10. It may be beyond of the scope of the current manuscript, but I would like to see the effect of Rad23 in the reconstitution system.
4. Related to this, does Rad23 UBL have an inhibitory effect in the competition experiments in Figure 5?
5. Because it is a bit complicated for readers, it would be better to present the summary illustration as the main figure.

Minor comments:

1. Each ubiquitin-binding domain of the proteasomal ubiquitin receptors recognizes the hydrophobic patch of monomeric ubiquitin, respectively. The degree of degradation of different ubiquitin chain types may simply depend on the distance between the ATPase entry port and the distal ubiquitin molecules of the chains. Is it possible to calculate the physical lengths of each ubiquitin chains from known structural data and discuss it?
2. It would be better to present the data of SDS-PAGE and fluorescent images to show the purity of all the ubiquitinated proteins.
3. Page 9, l. 6-8. What does it mean that the C-term K48 chain did not induce degradation? Some explanation would be included.
4. Table 2. Ub4(lin)-GFP-35 should be correct.
5. Page 17, l. 7. Table 2 would be correct.
6. Page 17, l. 9. Again, Figure 7a and Table 2 would be correct.

Reviewer #2 (Remarks to the Author):

In the submitted manuscript the authors aim to explain how three proteasomal ubiquitin receptors (Rpn10, Rpn13 and Rpn1) provide a versatile recognition platform for various types of ubiquitinated substrates.

The study is entirely based on the generation of various defined GFP-based substrates and on using purified reconstituted 26S proteasome (wild-type or mutants of various ubiquitin receptors).

The manuscript is clearly written and helps explain how 26S proteasome recognizes various types

of presented ubiquitinated substrates.

There are few points that should be further addressed:

Figure 3. Multiple studies have shown that purified 26S proteasomes can degrade K63 ubiquitin chains. Nevertheless, many studies (10.1016/j.cell.2009.01.041, 10.1038/emboj.2012.354) show that in vivo K63 chains do not provide proteasomal degradation signal. The authors should comment on that and provide evidence that in vivo 26S proteasomes could indeed degrade K63-linked substrates.

Figure 5. What is the purpose of analyzing linear ubiquitin chain degradation in yeast? The presence of E3 ligase LUBAC (or any other ligase able to generate linear ubiquitin chains), as well as presence of enzymatically assembled linear ubiquitin chains has not been shown in yeast at all. Also, inserting a short linker (GSGGGG) between the ubiquitin domains to turn it into proteasomal target is physiologically irrelevant, as it does not mimick any ubiquitin linkage whatsoever.

Figure 5. What about Ub1-GFP-35 and 35-GFP-Ub1 affinities in the context of the proteasome?

Figure 5. The authors show that protein affinity to the proteasome by itself does not predict degradation. What is the actual proof for their claim that "placement of the substrate on the proteasome affects degradation either by determining how well the proteasome is able to engage the protein at the initiation site or by inducing the proteasome to take up more (or less) active conformation?"

The authors should show how Ubp6-deficient mutant of Rpn1, either alone or together with Ub-binding deficient mutant of Rpn1, could degrade UBL-GFP-95 substrate, to further strengthen their hypothesis.

Reviewer #3 (Remarks to the Author):

This is an elegant biochemical work that investigates important and still unresolved question role of the role of three different ubiquitin receptors, Rpn10, Rpn13 and Rpn1, in protein degradation by proteasomes. Specifically, authors focus on the question of whether receptor requirement depends on the type of the ubiquitin chain on the substrate. The authors have created multiple model substrates carrying different types of ubiquitin chains, and have engineered proteasomes lacking one or two out of three receptors. Their most important findings are that Rpn10 appear to be sufficient for degradation of substrates marked by K48 chains (Fig. 2), while 2 receptors are required for K63 chains. They also found that Rpn13 slows degradation of K48 chain substrates. Although results are important, it is unclear how will they translate in vivo. Dr. Finley is well known for elegant genetic experiments, and I surprised that there are none in this paper.

Another problem of the study is that authors did not use their biochemical system to its full potential. They assay degradation by following fluorescence decrease of the model GFP-derived substrate. However, fluorescence never goes to zero, and 20-60% of substrate remains undegraded. Is it because substrate does not bind by the proteasome, or is it because proteasome releases it, perhaps after deubiquitylation?

We thank the reviewers for their efforts on our behalf and their constructive comments. We have addressed all the reviewers' concerns explicitly and performed a substantial number of additional experiments.

Reviewer #1 (Remarks to the Author):

In the current manuscript entitled "The Proteasome 19S Cap and its Three Identified Ubiquitin Receptors Provide a Versatile Recognition Platform for Substrates", Matouschek and colleagues define the molecular mechanism of how proteasomal ubiquitin receptors discriminate ubiquitin chains of different topologies using fully reconstituted system. In this well designed and carefully prepared manuscript, the authors clearly demonstrate that Rpn10 is the primary Ub receptor of the proteasome and that Rpn13 has an inhibitory role for proteasomal degradation. Moreover, the authors redefine the role of Rpn1 and Rpn13 as UBL receptors. Overall, I found this manuscript to be a fascinating read that describes very important research in future proteasome studies. However, a few issues should be addressed to strengthen the several of the conclusions.

Major points:

1. Throughout the study, proteasomal degradation was only analyzed by monitoring fluorescence loss of GFP. Are the substrate proteins truly degraded? Also, Ub5(K11) substrate and Ub5(M1) substrate were not degraded at all, but considering that these type of ubiquitin chains can nonselectively bind to the proteasome, they may be deubiquitinated. It is necessary to confirm the proteasomal degradation/deubiquitination by western blot analysis at least some timepoints of the unfolding assay.

To investigate the issue raised by the reviewer, we tested whether substrates are degraded or deubiquitinated by Western blotting against GFP. We found that substrates targeted to the proteasome by ubiquitin chains linked through Lys48 are degraded so that GFP is no longer detected by anti GFP antibodies (Figure S5). In contrast substrates with ubiquitin chains linked through Lys11 or M1, which do not show a loss of fluorescence during incubation with the proteasome, persist as ubiquitinated proteins during the incubation with proteasome (Figure S5). Thus, these proteins appear to be neither unfolded nor deubiquitinated.

2. There is no data for degradation assay of monomeric Ub-fused GFP-tail. The authors only presented the data of GFP-35 in Figure S3. Is Ub-GFP-tail not degraded at all? If Ub-GFP-tail is degraded by the proteasome, K11 and M1-linked ubiquitin chains have an inhibiting role for degradation.

We performed these experiments and found that Ub-GFP-35 but not 35-GFP-Ub degraded measurably but weakly (Figure R1a). This degradation is due to residual binding of Ub-GFP-35 to known ubiquitin receptors because it is inhibited when these are attenuated by mutation (Figure R1b). Thus, as the reviewer suspected, the formation of linear (M1-linked) and K11-linked ubiquitin chains does inhibit the degradation of mono-ubiquitin fusion protein. Whether this effect is relevant physiologically is beyond the scope of this manuscript.

Given this new observation, we tested whether Ub₅(K63)-GFP-35 degradation is due to the residual interaction of the mono-ubiquitin fusion protein with the proteasome. We found buffer conditions that reduced degradation of Ub-GFP-35 almost to background levels (Figure R1b). Ub₅(K48)-GFP-35 and Ub₅(K63)-GFP-35 are degraded by proteasome under these conditions just as before (Figure R2) showing that K63-linked ubiquitin chains do indeed support proteasomal degradation robustly.

3. The authors discuss that Rpn10 is the main Ub receptor and that Rpn1 and Rpn13 mainly function as UBL receptors. So, it is assumed that Rad23 binds to Rpn1 or Rpn13 and passes the Rad23-bound K48Ub substrate to Rpn10. It may be beyond of the scope of the current manuscript, but I would like to see the effect of Rad23 in the reconstitution system.

We agree that these are interesting concepts to explore but, as the reviewer suggests, the experiments are outside the scope of the manuscript. It will be a nontrivial set of experiments given the multiplicity of receptors and the existence of receptor-receptor interactions. Nevertheless, we will prioritize this investigation and ideally it will be the theme of our next paper.

4. Related to this, does Rad23 UBL have an inhibitory effect in the competition experiments in Figure 5?

The Rad23 UBL domain inhibits degradation of UBL-GFP-tail (Figure S8) but not of substrates with K48-linked polyubiquitin chains (Figure R3a, b). Interestingly, adding free UBL domain to degradation reactions of Ub₄(lin) substrates has a complex effect, inhibiting degradation at low concentrations and stimulating degradation high concentrations (Figure R3c). This suggests that the binding sites for Ub₄(lin) and UBL domains overlap. Stimulation of degradation by UBL domains has recently been described (Kim, H.T. & Goldberg A.L. (2018) Proc. Natl. Acad. Sci. USA **115**, E11642-E11650).

5. Because it is a bit complicated for readers, it would be better to present the summary illustration as the main figure.

We have included Figure S5 in the main manuscript as a new Figure 4.

Minor comments:

1. Each ubiquitin-binding domain of the proteasomal ubiquitin receptors recognizes the hydrophobic patch of monomeric ubiquitin, respectively. The degree of degradation of different ubiquitin chain types may simply depend on the distance between the ATPase entry port and the distal ubiquitin molecules of the chains. Is it possible to calculate the physical lengths of each ubiquitin chains from known structural data and discuss it?

The ubiquitin binding sites on Rpn10 and Rpn13 within the proteasome complex are not well defined in the current structures, which is thought to be largely due to inherent flexibility, and the receptor of the initiation region on the proteasome has not been established firmly. Therefore, we prefer to cite only estimates approximately distances as we currently do in Figure 1.

2. It would be better to present the data of SDS-PAGE and fluorescent images to show the purity of all the ubiquitinated proteins.

Some of the substrates have been described previously (Martinez-Fonts, K. and Matouschek, A. (2016). A Rapid and Versatile Method for Generating Proteins with Defined Ubiquitin Chains. *Biochemistry* **55**, 1898-1908; cited as [24]). For all the substrates not described previously, we now show samples analyzed by SDS-PAGE in a new Figure S1.

3. Page 9, l. 6-8. What does it mean that the C-term K48 chain did not induce degradation? Some explanation would be included.

K48-linked ubiquitin chains target our substrate more efficiently for degradation when placed at the N-terminus than when placed at the C-terminus of our test substrate. The C-terminal K48 chain does target the protein for degradation by WT and Rpn10 proteasome but less efficiently than the N-terminal chain. The substrate with a C-terminal chain is not degraded by Rpn13, Rpn1, and TM proteasome, whereas substrate with an N-terminal chain is degraded somewhat. A possible explanation is that binding of the K48-linked chain positions the substrate on the proteasome in a manner that makes it difficult for the proteasome to engage the initiation region. The ubiquitin binding domain of Rpn10 is connected to the proteasome through a flexible linker whereas the ubiquitin binding domain of Rpn13 is connected by a short linker, and the ubiquitin binding site of Rpn1 is directly on the proteasome surface. Thus, substrate bound to Rpn10 is able to explore more space to allow the initiation region to reach its receptor than substrate bound to Rpn13 or Rpn1.

4. Table 2. Ub4(lin)-GFP-35 should be correct.

Fixed.

5. Page 17, l. 7. Table 2 would be correct.

Fixed

6. Page 17, l. 9. Again, Figure 7a and Table 2 would be correct.

Fixed

Reviewer #2 (Remarks to the Author):

In the submitted manuscript the authors aim to explain how three proteasomal ubiquitin receptors (Rpn10, Rpn13 and Rpn1) provide a versatile recognition platform for various types of ubiquitinated substrates.

The study is entirely based on the generation of various defined GFP-based substrates and on using purified reconstituted 26S proteasome (wild-type or mutants of various ubiquitin receptors).

The manuscript is clearly written and helps explain how 26S proteasome recognizes various types of presented ubiquitinated substrates.

There are few points that should be further addressed:

Figure 3. Multiple studies have shown that purified 26S proteasomes can degrade K63 ubiquitin chains. Nevertheless, many studies (10.1016/j.cell.2009.01.041, 10.1038/emboj.2012.354) show that in vivo K63 chains do not provide proteasomal degradation signal. The authors should comment on that and provide evidence that in vivo 26S proteasomes could indeed degrade K63-linked substrates.

The reviewer raises an important point and we now mention this explicitly when introducing these experiments in the Results section, with the two references mentioned by the reviewer. (The manuscript had already cited the two studies in several places.)

Since the manuscript is focused on the biochemical characterization of the proteasomal degradation of defined substrate in vitro, we propose that it is beyond its scope to investigate degradation of substrates with K63-linked chains in vivo. There is already evidence that K63 ubiquitin chains can target proteins to proteasomal degradation in yeast. Mga2-p120 is ubiquitinated by Rsp5 and a high level of K63-linked chains are found in ubiquitinated Mga2-p120 purified from yeast. In addition, many proteins found bound to the proteasome contain K63-linked chains (Saeki, Y., Kudo, T., Sone, T., Kikuchi, Y., Yokosawa, H., Toh-e, A., Tanaka, K. (2009). Lysine 63-linked polyubiquitin chain may serve as a targeting signal for the 26S proteasome. *EMBO J.* **28**, 359-371. The paper is cited in the manuscript as [23]). Tsuchiya *et al.* (In Vivo Ubiquitin Linkage-type Analysis Reveals that the Cdc48-Rad23/Dsk2 Axis Contributes to K48-Linked Chain Specificity of the Proteasome. *Mol Cell* **66**, 488-502 e487 (2017), cited as [28]) come to similar conclusions following different lines of investigation.

Figure 5. What is the purpose of analyzing linear ubiquitin chain degradation in yeast? The presence of E3 ligase LUBAC (or any other ligase able to generate linear ubiquitin chains), as well as presence of enzymatically assembled linear ubiquitin chains has not been shown in yeast at all. Also, inserting a short linker (GSGGGG) between the ubiquitin domains to turn it into proteasomal target is physiologically irrelevant, as it does not mimick any ubiquitin linkage whatsoever.

We performed degradation reactions with proteasome purified from mouse embryonic fibroblasts by affinity chromatography using a FLAG-tag attached to the C-terminus of Rpn11 in parallel with proteasome purified from yeast, also using a FLAG-tag attached to the C-terminus of Rpn11. The degradation reactions with the two proteasomes yield the same degradation patterns (new Figure S6). In particular, substrate modified by a ubiquitin chain linked through Lys11 is not degraded by mouse proteasome, while substrate with a K48-linked chain, and to a lesser extent substrate with a K63-linked chain, is degraded. The data are now mentioned in the manuscript and shown in a new Figure S6.

Yes, ubiquitin chains in which domains are connected by short linkers do not occur naturally. We included these observations in the manuscript because these ubiquitin chains provide a convenient method to create reactive proteasome substrates and now mention this point in the manuscript explicitly. We are also glad to remove these data from the manuscript if doing so improves readability.

Figure 5. What about Ub1-GFP-35 and 35-GFP-Ub1 affinities in the context of the proteasome?

Please see our response to Reviewer 1, major point 2, where we discuss this point in some detail.

Figure 5. The authors show that protein affinity to the proteasome by itself does not predict degradation. What is the actual proof for their claim that “placement of the substrate on the proteasome affects degradation either by determining how well the proteasome is able to engage the protein at the initiation site or by inducing the proteasome to take up more (or less) active conformation?”

The reviewer is correct - the statement proposes a possible mechanism that could explain the experimental observations and therefore is speculative. We now make this clear in the wording of the sentence.

The authors should show how Ubp6-deficient mutant of Rpn1, either alone or together with Ub-binding deficient mutant of Rpn1, could degrade UBL-GFP-95 substrate, to further strengthen their hypothesis.

We find that mutating the binding site for the UBL-domain of Ubp6 (the T2 site) does not abolish degradation of UBL-GFP-95. The protein is degraded robustly by proteasome in which all known ubiquitin receptors as well as the T2 (quadruple mutant or QM proteasome). We now show these data in a new Figure S9 and discuss them in the manuscript at the appropriate point.

We also asked whether the unexpectedly robust degradation of Ub₅(K48)-GFP-35 by TM proteasome was due to the T2 site. This is not the case as this protein degraded by TM and QM proteasome equally well (Figure R4).

Reviewer #3 (Remarks to the Author):

This is an elegant biochemical work that investigates important and still unresolved question role of the role of three different ubiquitin receptors, Rpn10, Rpn13 and Rpn1, in protein degradation by proteasomes. Specifically, authors focus on the question of whether receptor requirement depends on the type of the ubiquitin chain on the substrate. The authors have created multiple model substrates carrying different types of ubiquitin chains, and have engineered proteasomes lacking one or two out of three receptors. Their most important findings are that Rpn10 appear to be sufficient for degradation of substrates marked by K48 chains (Fig. 2), while 2 receptors are required for K63 chains. They also found that Rpn13 slows degradation of K48 chain substrates. Although results are important, it is unclear how will they translate in vivo. Dr. Finley is well known for elegant genetic experiments, and I surprised that there are none in this paper.

Another problem of the study is that authors did not use their biochemical system to its full potential. They assay degradation by following fluorescence decrease of the model GFP-derived substrate. However, fluorescence never goes to zero, and 20-60% of substrate remains undegraded. Is it because substrate does not bind by the proteasome, or is it because proteasome releases it, perhaps after deubiquitylation?

The reviewer is correct, and the degradation reactions do not proceed to 100% of the possible extent. This behavior is not uncommon for complex biochemical reactions reconstituted in vitro. In our case, the explanation is most likely that substrate protein slowly inactivates, presumably by binding non-productively to the proteasome. Some but not all of the substrate remaining at the end of the degradation reaction retains its ubiquitin modification, suggesting that deubiquitination is at least not the only process inhibiting degradation (Figure R5).

Figure R1: Degradation of Ub-GFP-35 and 35-GFP-Ub.

(a) Degradation of Ub-GFP-35 and 35-GFP-Ub by WT proteasome under standard conditions (no proteasome shows Ub-GFP-35 in the degradation reaction without proteasome).

(b) Degradation of Ub-GFP-35 by WT and quadruple mutant (QM) proteasome in standard and modified buffer indicated by * (50 mM TrisHCl pH 7.6, 5 mM MgCl₂, 50 mM NaCl, 1% glycerol, 2 mg/mL BSA, 1 mM ATP, 10 mM creatine phosphate, 0.1 mg/mL creatine phosphokinase, 4 mM DTT). We created QM proteasome by mutating the binding site for the divergent UBL domain of Ubp6 (also called the T2 binding site) in addition to the ubiquitin binding sites in Rpn10 and Rpn13, and the UBL binding site on Rpn1 (T1).

Figure R2: Degradation of Ub₅(K48)-GFP-35 and Ub₅(K63)-GFP-35.

(a) Degradation of Ub₅(K48)-GFP-35 and Ub₅(K63)-GFP-35 by WT proteasome under standard conditions and in modified buffer (indicated by *).

(b) Degradation of Ub₅(K63)-GFP-35 by the indicated proteasome in modified buffer.

Figure R3: Effect of the UBL domain of Rad23 on protein degradation. Effect of increasing concentration of UBL on the degradation of Ub₅(K48)-GFP-35 by WT proteasome **(a)** or Rpn10 proteasome **(b)**, and on the degradation of Ub₄(lin)-GFP-35 by Rpn13 proteasome **(c)**. Compare to Figure 6.

Figure R4: Degradation of Ub₅(K48)-GFP-35 by WT, TM and QM proteasome

α - Ub

Figure R5: Deubiquitination of Ub₅(K48)-GFP-35 during proteasomal degradation. Anti-ubiquitin Western blot of substrate at the starting and end points of a standard degradation reaction.

REVIEWERS' COMMENTS:

Reviewer #1 (Remarks to the Author):

The authors have addressed my major concerns. I'm impressed with the amount of work done including the mammalian proteasome experiments. This is a significant study and deserves publication in Nature Communications.

Reviewer #2 (Remarks to the Author):

The authors have successfully addressed all of my questions and raised issues. Unless addressed experimentally, my concerns were explained and included, together with appropriate references, into the manuscript. I therefore have no more comments or complaints regarding the current version of the manuscript and recommend it for publication.